# T2AV-Compass: Towards Unified Evaluation for Text-to-Audio-Video Generation

**Zhe Cao** [* 1]   **Tao Wang** [* 1]   **Jiaming Wang** [* 1]   **Yanghai Wang** [* 1]
**Yuanxing Zhang** [2]   **Jiahao Wang** [1]   **Jialu Chen** [2]
**Miao Deng** [1]   **Chenxi Liao** [1]   **Yize Zhang** [1]   **Yubin Guo** [1]   **Zhaoxiang Zhang** [3]   **Jiaheng Liu** [1 †]

## Abstract

Text-to-Audio-Video (T2AV) generation aims to synthesize temporally coherent video and semantically synchronized audio from natural language, yet its evaluation remains fragmented, often relying on unimodal metrics or narrowly scoped benchmarks that fail to capture cross-modal alignment, instruction following, and perceptual realism under complex prompts. To address this limitation, we present T2AV-Compass, a unified benchmark for comprehensive evaluation of T2AV systems, consisting of 500 diverse and complex prompts constructed via a taxonomy-driven pipeline to ensure semantic richness and physical plausibility. T2AV-Compass introduces a dual-level evaluation framework that combines objective signal-level metrics for video quality, audio quality, and cross-modal alignment with judge-based diagnostics for instruction following and realism assessment. Extensive evaluation of 15 representative T2AV systems reveals that even the strongest models fall substantially short of human-level realism and cross-modal consistency, with persistent failures in audio realism, fine-grained synchronization, and instruction following, etc. These results highlight the value of T2AV-Compass as a challenging and diagnostic testbed for advancing T2AV generation.

## 1. Introduction

Generative AI has witnessed a paradigm shift from unimodal synthesis to cohesive multimodal content creation (Singer

---

[*]Equal contribution   [†] Corresponding author.  [1]Nanjing University, Nanjing, China [2]Kling Team, Kuaishou Technology, Beijing, China [3]Institute of Automation, Chinese Academy of Sciences, Beijing, China. Correspondence to: Jiaheng Liu <liujiaheng@nju.edu.cn>.

*Proceedings of the 43rd International Conference on Machine Learning*, Seoul, South Korea. PMLR 306, 2026. Copyright 2026 by the author(s).

et al., 2023; Ho et al., 2022; Guo et al., 2024; Wang et al., 2025c; Tang et al., 2025), with Text-to-Audio-Video (T2AV) generation emerging as a frontier that unifies visual dynamics and auditory realism. Recent breakthroughs, from proprietary systems like Sora (OpenAI, 2024) and Veo (DeepMind, 2024) to open research efforts (Yang et al., 2025b; Ruan et al., 2023; Lin et al., 2023), have demonstrated the ability to generate high-fidelity audio-video from textual prompts. Despite rapid progress, **the evaluation of T2AV systems remains fundamentally underdeveloped.**

Existing benchmarks largely evolve from unimodal or weakly multimodal settings. On the one hand, existing benchmarks either prioritize visual quality in isolation (e.g., VBench (Huang et al., 2024), EvalCrafter (Liu et al., 2024b) or focus solely on audio fidelity (e.g., AudioCaps (Kim et al., 2019), AudioLDM-Eval (Liu et al., 2024a)), failing to capture the cross-modal semantic alignment and temporal synchronization that define realistic T2AV generation. On the other hand, emerging audio–video benchmarks take important steps toward joint evaluation, yet they often face critical trade-offs: limited coverage of fine-grained coupling phenomena, insufficient handling of long and compositional prompts, reliance on narrow metric sets, or a lack of interpretable diagnostic signals (e.g., instruction following, realism). For example, current evaluations struggle to answer core questions: Do generated sounds correspond to visible events? Are multiple audio sources synchronized with complex visual interactions? Does the model faithfully follow detailed instructions while maintaining physical and perceptual realism? These challenges are exacerbated by the intrinsic complexity of T2AV generation. Specifically, high-quality output requires simultaneous success along multiple axes: unimodal perceptual quality, cross-modal semantic alignment, precise temporal synchronization, instruction following under compositional constraints, and realism grounded in physical and commonsense knowledge.

To address this gap, in Figure 1, we introduce **T2AV-Compass**, the first comprehensive benchmark designed specifically for evaluating text-to-audio-video generation. Specifically, first, T2AV-Compass employs a taxonomy-driven curation pipeline to construct 500 complex prompts

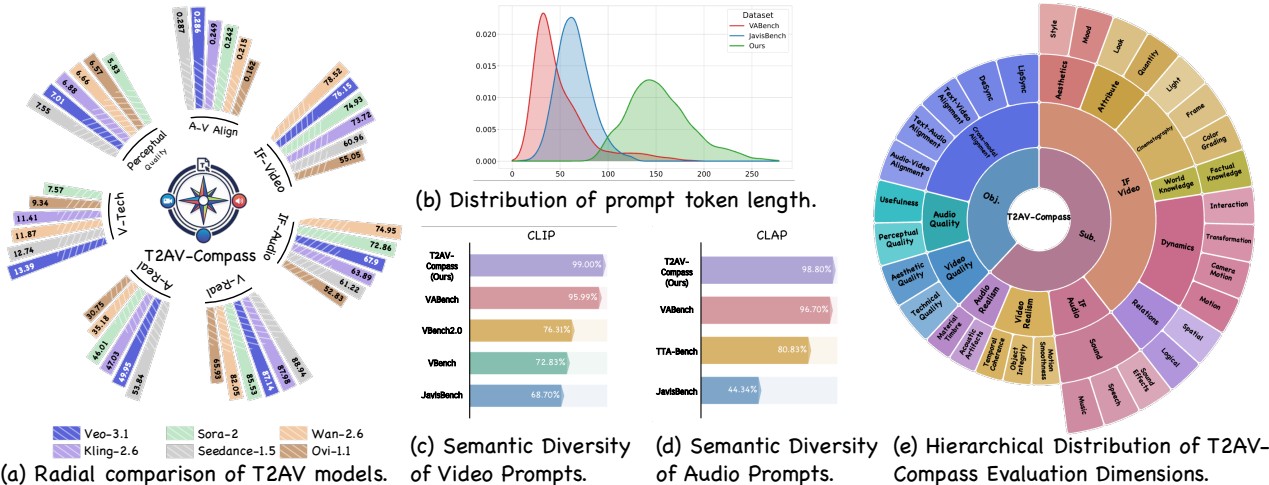

*Figure 1.* **Overview of T2AV-Compass analysis and evaluation taxonomy.** (a) Radial comparison of representative T2AV models under our evaluation suite. (b) Prompt token-length distribution. (c–d) Semantic diversity of video/audio prompts quantified via embedding similarity (higher indicates broader coverage). (e) Hierarchical distribution of evaluation dimensions, clearly organizing objective metrics and MLLM-based assessments across video, audio, and cross-modal alignment.

with broad semantic coverage and challenging audiovisual scenarios, which impose precise constraints across cinematography, physical causality, and acoustic environments. Second, we propose a dual-level evaluation framework that integrates objective evaluation based on classical automated metrics with subjective evaluation based on MLLM-as-judge. The objective evaluation quantifies video quality (technical fidelity, aesthetic appeal), audio quality (acoustic realism, semantic usefulness), and cross-modal alignment (text-audio/video semantic consistency, temporal synchronization). The subjective evaluation mainly evaluates video and audio instruction following abilities based on well-defined checklists and perceptual realism (e.g., physical plausibility and fine-grained details), which aims to address the limitations of automated metrics in capturing nuanced semantic and causal coherence. Code and data are available at NJU-LINK/T2AV-Compass and HuggingFace.

In summary, our contributions are threefold as follows:

- **Taxonomy-Driven High-Complexity Benchmark:** We introduce *T2AV-Compass*, a benchmark comprising 500 dense prompts synthesized through a hybrid pipeline of taxonomy-based curation and video inversion. It targets fine-grained audiovisual constraints— such as off-screen sound and physical causality— frequently overlooked in existing evaluations.

- **Unified Dual-Level Evaluation Framework:** We propose a paradigm that integrates objective signal metrics with a novel *MLLM-as-a-Judge* protocol. Based on well-designed QA checklists, our framework bridges the gap between low-level fidelity and high-level semantic logic with enhanced interpretability.

- **Extensive Benchmarking and Empirical Insights:** We conduct a systematic evaluation of 15 state-of-the-art T2AV systems, including leading proprietary models like Veo-3.1 and Kling-2.6. Our analysis unveils a critical "Audio Realism Bottleneck", revealing that current models struggle to synthesize physically grounded audio textures that match the visual fidelities.

## 2. Related Work

**Benchmarks for Single-Output-Modality Generation** Early video generation benchmarks focused on visual fidelity and text–video relevance (Huang et al., 2024; Liu et al., 2024b; 2023; He et al., 2024; Tong et al., 2026; Zheng et al., 2025; Duan et al., 2025; Guo et al., 2025), while recent work extends evaluation to audio and audio–video generation, emphasizing temporal alignment and semantic controllability (Wang et al., 2026; Hua et al., 2025; Iashin et al., 2024; Li et al., 2024). For instance, VBench assesses visual quality and text–video alignment (Huang et al., 2024), and TTA-Bench evaluates perceptual quality via human annotations (Wang et al., 2026). However, unimodal metrics remain insufficient for capturing audio–video consistency in timing, spatial cues, and semantics (Iashin et al., 2024; Li et al., 2024; Hua et al., 2025).

**Emerging text-to-audio-video generation benchmarks.** In Table 1, recent efforts introduce evaluation sets for joint audio-video generation. JavisBench focuses on diverse open-domain audio–video generation and spatio-temporal alignment stress tests (Liu et al., 2025), while Verse-Bench and Harmony-Bench probe synchronized generation across different acoustic scenarios (Wang et al., 2025a; Hu et al., 2025). VABench is a related contemporaneous benchmark

*Table 1.* **Comparison of representative generative benchmarks.** We provide a detailed comparison of multiple benchmarks in following dimensions: **Avg Tokens/Sub./Events.**: **Avg Tokens** are calculated using the Qwen3 tokenizer(Yang et al., 2025a). **Sub.** refers to the average of distinct themes or subjects addressed in the benchmark dataset. **Events** indicates the number of events within each subject that are considered for evaluation. Additionally, the table includes a breakdown of sound types in terms of: `Sound` (general sound), `Music` (musical content), `Speech` (speech-related content), where applicable. The evaluation dimensions include: `VQ` (Video Quality), `AQ` (Audio Quality), `CMA` (Cross-Modal Alignment), `IF` (Instruction Following, which includes tasks involving constraints), and `RE` (Realism Fidelity, which assesses the perceived accuracy of generated content beyond mere signal-level quality).

| Benchmark | Task | Items | #Metrics | Avg Tokens./Sub./Events. | Sound Types | Eval. Dimensions |
|---|---|---|---|---|---|---|
| VBench (Huang et al., 2024) | T2V | 946 | 16 | 10/1.34/1.06 | - | `VQ` |
| TTA-Bench (Wang et al., 2026) | T2A | 2,999 | 10 | 20/2.86/1.68 | `Sound` `Music` `Speech` | `AQ` |
| JavisBench (Liu et al., 2025) | T2AV | 10,140 | 5 | 65/3.68/1.78 | `Sound` | `VQ` `AQ` `CMA` |
| Verse-Bench (Wang et al., 2025a) | TI2AV | 600 | 4 | 68/2.01/1.38 | `Sound` `Speech` | `VQ` `AQ` `CMA` |
| Harmony-Bench (Hu et al., 2025) | TI2AV | 150 | 6 | 🔒 | `Sound` `Speech` | `VQ` `AQ` `CMA` |
| UniAVGen (Zhang et al., 2025) | TIA2V | 100 | 3 | 🔒 | `Speech` | `VQ` `AQ` `CMA` |
| VABench (Hua et al., 2025) | T2AV & I2AV | 778 | 15 | 50/3.01/2.31 | `Sound` `Music` `Speech` | `VQ` `AQ` `CMA` |
| **T2AV-Compass (Ours)** | **T2AV** | **500** | **13** | **154 / 4.03 / 3.61** | `Sound` `Music` `Speech` | `VQ` `AQ` `CMA` `IF` `RE` |

that combines expert metrics with MLLM-based evaluation across T2AV, I2AV, and stereo audiovisual settings (Hua et al., 2025). Nevertheless, existing benchmarks are often limited in fine-grained taxonomies and evaluation metrics. These limitations motivate the development of T2AV-Compass (Huang et al., 2024; Liu et al., 2024b; 2023; Tong et al., 2026; He et al., 2024; Zheng et al., 2025; Wang et al., 2026; Hua et al., 2025).

## 3. T2AV-Compass

### 3.1. Data Construction

To ensure the diversity and complexity of the dataset, we employ a three-stage construction pipeline comprising taxonomy-based prompt design, multi-source data collection, and real-world video inversion in Figure 2.

**Data Collection** To establish broad semantic coverage, we aggregate prompts from high-quality sources, including VidProM, the Kling AI community, LMArena, and Shot2Story (Wang & Yang, 2024; Kuaishou Technology, 2024; LMArena Community, 2024; Han et al., 2025); details are provided in Appendix H. We encode all prompts using all-mpnet-base-v2, deduplicate with a cosine-similarity threshold of 0.8 (Reimers & Gurevych, 2019). We then apply square-root sampling to preserve semantic distinctiveness while preventing the dominance of frequent topics.

**Prompt Refinement and Alignment.** Raw prompts often lack the descriptive density needed to stress-test state-of-the-art models (e.g., Veo 3.1, Sora 2, Kling 2.6) (DeepMind, 2024; OpenAI, 2024; Kuaishou Technology, 2024; Wang et al., 2025b). To address this, we employ Gemini-2.5-Pro rewriting sampled prompts to add visual, motion, acoustic, and cinematographic constraints, increasing the average length from 54 to about 154 tokens and the average number of constraint points from roughly 5 to 10. Human refinement

then removes non-compliant, overly long, or illogical cases, reducing candidates to 400 complex high-quality rewritten prompts.

**Real-world Video Inversion.** To counterbalance potential hallucinations in text-only generation and ensure physical plausibility (Guo et al., 2025; Duan et al., 2025), we introduce a Video-to-Text inversion stream. We select 100 diverse, high-fidelity video clips (4–10s) from YouTube and utilize Gemini-2.5-Pro to generate dense, temporally aligned captions. Discrepancies between the generated prompts and the source ground truth are resolved via human-in-the-loop verification, yielding 100 high-quality prompts anchored in real-world dynamics for reliable evaluation.

### 3.2. Dataset Statistics

**Distribution and Diversity.** As depicted in Figure 1(b), our prompts exhibit notably higher token counts compared to existing baselines (e.g., JavisBench, VABench), more accurately mirroring the complexity of real-world user queries. The dataset encompasses a broad spectrum of themes, soundscapes, and cinematographic styles, with the corresponding hierarchical QA distribution detailed in Figure 3. Figure 4 summarizes dataset statistics and category distributions. To quantify diversity, we analyze the semantic retention rates of CLIP (video) and CLAP (audio) embeddings after deduplication. As shown in Figures 1(c) and (d), our benchmark demonstrates superior semantic distinctiveness across both modalities, significantly outperforming concurrent datasets.

### 3.3. Dual-Level Evaluation Framework

We introduce a dual-level evaluation framework for T2AV generation designed to be both systematic and reproducible, as illustrated in Figure 1(e). At the objective level, we decompose system performance into three complementary pillars: (i) video quality, (ii) audio quality, and (iii) cross-modal

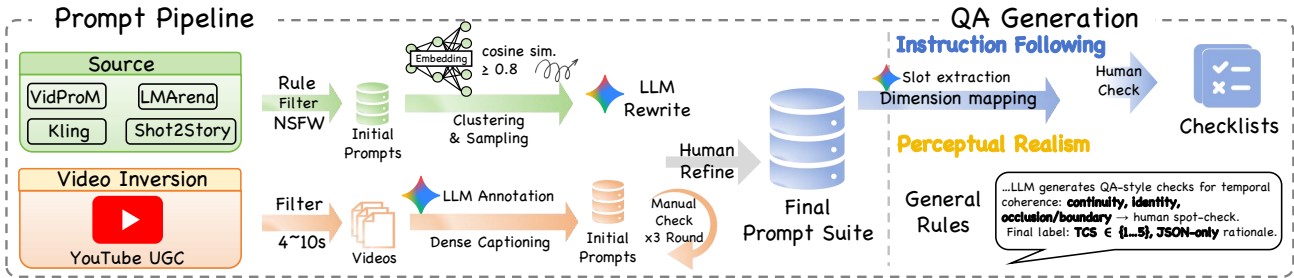

*Figure 2.* **Data construction and checklist-based evaluation generation.** The prompt suite is constructed from (1) curated community prompts with semantic deduplication (cos $\geq$ 0.8), clustering-based sampling, LLM rewriting, and human refinement, and (2) a video-inversion stream using filtered 4–10s YouTube clips with dense captioning and manual verification. The finalized prompts are then converted into two types of checklists: instruction-alignment checks via slot extraction and dimension mapping, and perceptual-realism checks for video/audio quality. See Appendix H for detailed dataset construction and Appendix I for related prompts.

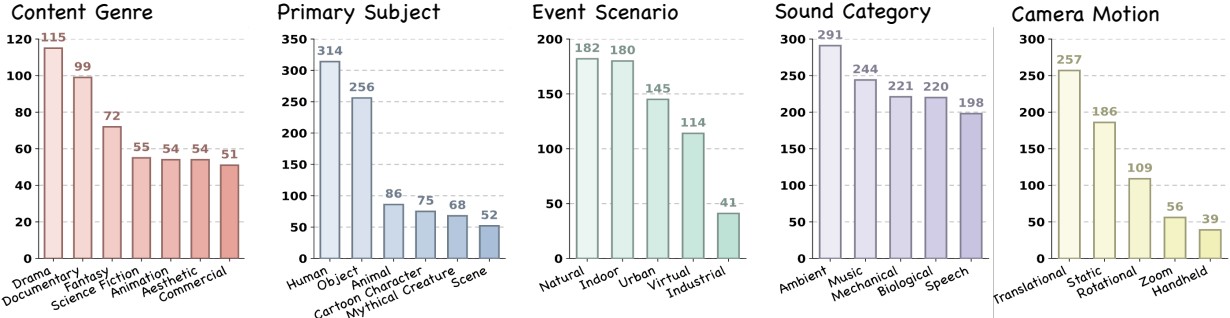

*Figure 3.* **Dataset statistics of T2AV-Compass.** Category distributions over five annotation dimensions.

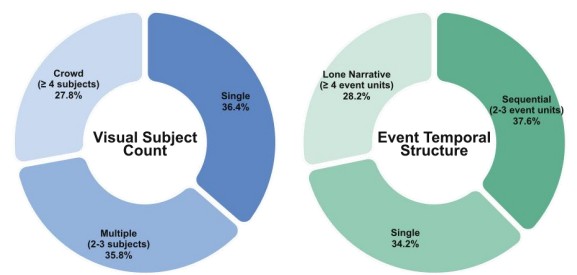

*Figure 4.* **Dataset statistics of T2AV-Compass.** Distributions of Visual Subject Count, Event Temporal Structure, see more audiovisual complexity factors in Figure 12 in Appendix G.

alignment. At the subjective level, we evaluate high-level semantic alignment through two dimensions: **Instruction Following (IF)** and **Realism**. We further analyze redundancy among metrics via a de-meaned correlation heatmap (Figure 10 in Appendix D.3). See Appendix I for MLLM-Judge prompts both on Instruction Following and Realism.

### 3.3.1. OBJECTIVE EVALUATION

We use a set of expert metrics to cover the three pillars above. Specifically, we measure video quality using perceptual and distributional metrics, audio quality using acoustic fidelity and intelligibility metrics, and cross-modal alignment using synchronization and semantic alignment metrics. Overall, these objective metrics offer a stable and comparable basis

for evaluating T2AV systems.

**Video Quality.** We evaluate the visual performance in low-level technical fidelity and high-level aesthetic appeal.

- **Video Technological Score (VT).** This metric quantifies low-level integrity, explicitly penalizing artifacts such as noise, blur, and compression distortions. We employ **DOVER++** (Wu et al., 2023) to score representative frames, aggregating frame-level predictions into a holistic video score. Higher VT values signify cleaner, sharper, and more photorealistic renderings.

- **Video Aesthetic Score (VA).** It captures high-level perceptual attributes, including composition, lighting, and color harmony. We utilize the **Aesthetic Predictor V2.5** (discus0434, 2024) on keyframes. By averaging these scores, VA serves as a proxy for subjective visual preference and artistic coherence.

**Audio Quality.** To isolate acoustic performance from cross-modal biases, we evaluate synthesized audio independent of the visual stream. Drawing on the evaluation framework of **Audiobox** (Vyas et al., 2023), we employ reference-free metrics to quantify low-level signal integrity and high-level semantic clarity. Furthermore, considering the specialized requirements for human-centric outputs, we

incorporate a dedicated analysis of speech quality to measure naturalness and intelligibility.

- **Audio Aesthetic Score(AA).** AA provides a holistic measure of acoustic excellence by balancing two key dimensions: Perceptual Quality (PQ), which assesses signal fidelity and timbre realism against artifacts, and Content Usefulness (CU), which evaluates the semantic density and the presence of identifiable auditory events. The AA score is defined as the arithmetic mean:

$$AA = \frac{PQ + CU}{2}.$$

- **Speech Quality Score(SQ).** For evaluating synthesized speech, we adopt the **NISQA** framework (Mittag et al., 2021). NISQA predicts subjective Mean Opinion Scores (MOS) directly from the synthetic signal, allowing for a robust characterization of speech naturalness, capturing nuanced degradations that traditional signal-to-noise metrics often overlook.

**Cross-modal Alignment.** We evaluate cross-modal alignment to ensure coherence across text, audio, and video. Our protocol assesses two dimensions: semantic consistency and temporal synchronization.

- **Text–Audio (T–A) Alignment.** We measure T–A alignment using **CLAP** (Elizalde et al., 2023), which maps text and audio into a shared embedding space. The cosine similarity between embeddings reflects semantic correspondence between the generated audio and the prompt, capturing fine-grained intent.

- **Text–Video (T–V) Alignment.** Visual adherence to the prompt is evaluated using **VideoCLIP-XL-V2** (Wang et al., 2024). We compute the cosine similarity between text and video feature embeddings to measure the high-level semantic consistency of the visual content in a unified space.

- **Audio–Video (A–V) Alignment.** To assess cross-modal consistency independent of text, we compute A–V semantic similarity via **ImageBind** (Girdhar et al., 2023). This score verifies if generated audio events align semantically with visual content.

- **Temporal Synchronization.** Beyond semantics, we assess temporal correspondence between audio and visual events using **DeSync (DS)** computed by Synchformer (Iashin et al., 2024). DS measures synchronization error as the absolute time offset between audio and visual onsets (lower is better). For talking-face scenarios, we additionally report lipsync using **LatentSync (LS)** (Li et al., 2024), a SyncNet-based lip-sync metric for diagnosing speech–lip synchronization.

### 3.3.2. SUBJECTIVE EVALUATION

To address the limitations of traditional metrics in capturing fine-grained semantic details and complex cross-modal dynamics, we establish a robust "MLLM-as-a-Judge" framework. This framework comprises two distinct evaluation tracks: Instruction Following (IF) and Realism. Crucially, we enforce a reasoning-first protocol, mandating that the judge explicitly articulates the rationale behind their decision prior to assigning a score on a 5-point scale. This protocol not only enhances interpretability but also significantly facilitates downstream error attribution. We provide qualitative examples in Figure 5 to illustrate typical failure modes under our checklist-based diagnosis. See Figure 13 and Figure 14 in Appendix J for more generated cases.

**Instruction Following (IF).** We first derive verifiable QA checklists from each prompt to instantiate abstract instructions into granular, measurable constraints. We employ Gemini-2.5-Pro as the judge to verify the generated video against these checklists at scale. In Table 10 and Table 11 of Appendix E, the taxonomy encompasses **7 primary dimensions** and **17 sub-dimensions**. The primary dimensions are as follows: (1). **Attribute:** Examines visual accuracy, focusing on Look and Quantity. (2). **Dynamics:** Assesses dynamic behaviors, including Motion, Interaction, Transformation, and Cam. Motion. (3). **Cinematography:** Scrutinizes directorial control, including Light, Frame, and Color Grading. (4). **Aesthetics:** Measures artistic integrity, decomposed into Style and Mood. (5). **Relations:** Verifies structural logic, evaluating Spatial and Logical connections. (6). **World Knowledge:** Tests grounding in reality, specifically Factual Knowledge of real-world scenarios. (7). **Sound:** Assesses the generation of auditory elements, covering Sound Effects, Speech, and Music.

**Realism.** While IF ensures the presence of prompt-specified content, it may overlook internal visual and audio inconsistencies or violations of physical laws. To bridge this gap, we introduce Realism metrics encompassing video and audio realism to scrutinize the physical and perceptual authenticity of the generated content. See Table 12 in Appendix E for detailed information.

- **Video Realism:** We evaluate visual plausibility through three complementary metrics: (1) **Motion Smoothness Score (MSS)**, which quantifies deviations from natural motion by penalizing jitter and temporal discontinuities; (2) **Object Integrity Score (OIS)**, which identifies anatomical distortions and transient visual artifacts; and (3) **Temporal Coherence Score (TCS)**, which assesses object permanence and the consistency of occlusion patterns over time.

- **Audio Realism:** Auditory fidelity is evaluated using

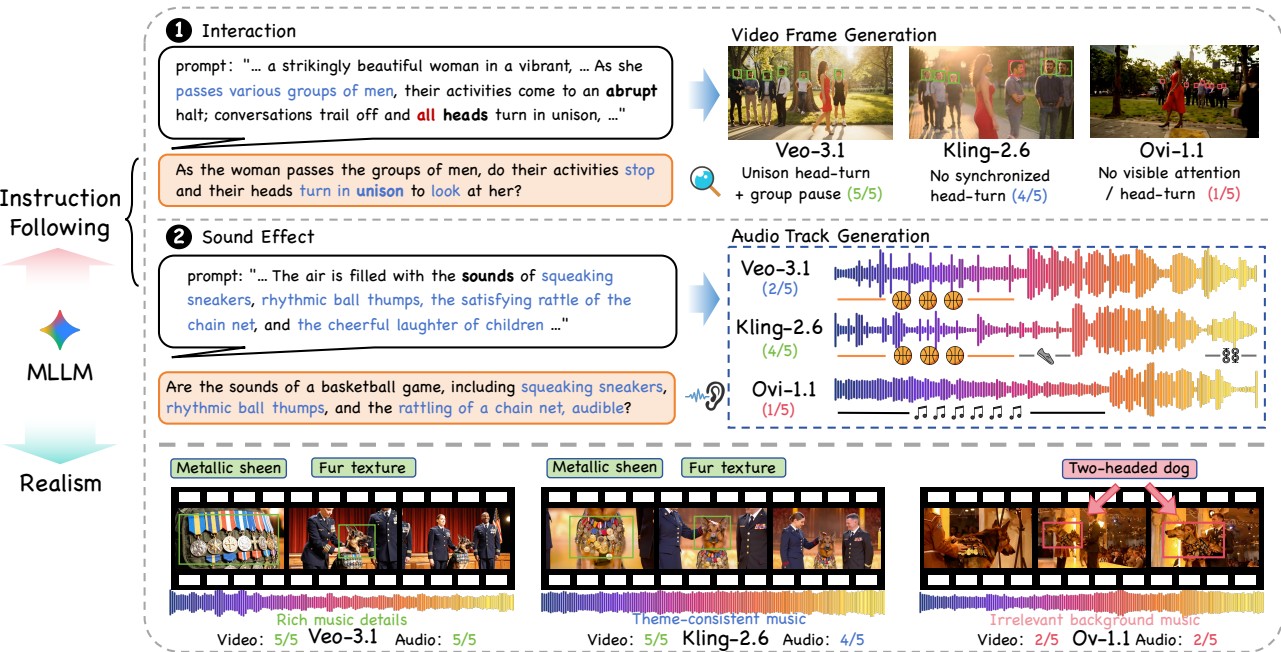

*Figure 5.* **Illustration of the subjective evaluation framework.** Our protocol provides interpretable diagnosis through two distinct tracks: *(Top)* Instruction following is evaluated via rigorous Q&A checklist pairs. *(Bottom) Realism* scrutinizes perceptual quality, rewarding fine-grained details while explicitly penalizing visual hallucinations or audio dissonance.

two perceptually grounded metrics: (1) **Acoustic Artifacts Score (AAS)**, which measures the presence of background noise and unnatural mechanical sounds that degrade immersion; and (2) **Material–Timbre Consistency (MTC)**, which verifies alignment between acoustic timbre and visible physical properties.

# 4. Experiments

## 4.1. Main Results

We evaluate 15 representative T2AV systems, comprising 7 closed-source end-to-end models, 3 open-source end-to-end models, and 5 composed generation pipelines: Veo-3.1 (DeepMind, 2024), Sora-2 (OpenAI, 2024), Kling-2.6 (Kuaishou Technology, 2024), Wan-2.6 and Wan-2.5 (Wan et al., 2025), Seedance-1.5 (Chen et al., 2025), PixVerse-V5.5 (Team, 2025), the open-source Ovi-1.1 (Low et al., 2025), LTX-2 (HaCohen et al., 2026) and Javis-DiT (Liu et al., 2025), and five modular pipelines Wan-2.2 + HunyuanVideo-Foley, Wan-2.2+MMAudio, HunyuanVideo-1.5 + HunyuanVideo-Foley, HunyuanVideo-1.5 (Wan et al., 2025; Shan et al., 2025; Cheng et al., 2025; Wu et al., 2025) + MMAudio and AudioLDM2 + MTV (Liu et al., 2024a; Weng et al., 2026). Please refer to D.2 for the specific analysis of each model. Table 2 and Table 3 present the objective and subjective results, respectively, and we have the following findings: (1) **Open vs. Closed-Source.** Closed-source models dominate the top of the leaderboard under the compact **Average** summary, which should not replace

dimension-level diagnosis, (Table 3), with **Veo-3.1** ranking first (70.29), followed by Sora-2 (69.83), Kling-2.6 (68.16), and Wan-2.6 (67.68). Among open-source end-to-end models, **LTX-2** is the strongest overall (63.72) and achieves the best **Video Realism** (89.95), while **Wan-2.6** leads **Instruction Following** (IF Video 78.52, IF Audio 74.95). Overall, the gap is most pronounced on high-level instruction-following and audio realism, whereas open-source and composed pipelines can be competitive on modality-specific realism. (2) **T2AV-Compass is Challenging.** No single model dominates all dimensions. For instance, while Veo-3.1 attains the highest overall average, it still shows major deficiencies in Audio Realism. (3) **Cascaded Pipelines are Strong but Disjoint.** Cascaded T2V → V2A systems can match end-to-end models on modality-specific quality (e.g., **Wan-2.2 + Hunyuan-Foley** reaches 89.63 Video Realism), but often lag in global audio-visual alignment due to fragmented optimization.

## 4.2. Further Analysis

**Where is the Bottleneck?** Figure 6 shows that **Dynamics** is the most challenging and discriminative dimension for video instruction following: even frontier models drop notably when prompts require complex motion execution and interactions, indicating a temporal-coherence bottleneck. On the audio side, our **IF Audio (IFA)** analysis in Figure 7 reveals uneven instruction following across sub-dimensions: **Sound Effects** is consistently the most error-

*Table 2.* Comparison of T2AV models across video quality, audio quality, and cross-modal alignment. A dash (–) indicates that the model is unable to generate human speech. Due to closed-souece security review, some video aren't generated successfully. See Table 5 in Appendix 6 for more detailed results, including per-sub-dimension scores.

| Method | Open-Source | Video Quality | | Audio Quality | | Cross-modal Alignment | | | | |
|---|---|---|---|---|---|---|---|---|---|---|
| | | VT↑ | VA↑ | AA↑ | SQ↑ | A-V↑ | T-A↑ | T-V↑ | DS↓ | LS↑ |
| **- T2AV** | | | | | | | | | | |
| Veo-3.1 | ✗ | 13.39 | 5.425 | 6.818 | 1.597 | 0.2856 | 0.2335 | 0.2438 | 0.6776 | 1.509 |
| Sora-2 | ✗ | 7.568 | 4.112 | 5.584 | 1.485 | 0.2419 | 0.2484 | 0.2432 | 0.8100 | 1.331 |
| Kling-2.6 | ✗ | 11.41 | 5.417 | 6.666 | 1.783 | 0.2495 | 0.2495 | 0.2449 | 0.7852 | 1.502 |
| Wan-2.6 | ✗ | 11.87 | 4.605 | 6.440 | 1.476 | 0.2149 | 0.2572 | 0.2451 | 0.8818 | 1.081 |
| Seedance-1.5 | ✗ | 12.74 | 5.007 | 7.403 | 1.766 | 0.2875 | 0.2320 | 0.2370 | 0.8650 | 1.560 |
| LTX-2 | ✓ | 7.160 | 4.661 | 6.742 | 1.597 | 0.1851 | 0.2365 | 0.2411 | 0.8756 | 1.339 |
| Wan-2.5 | ✗ | 13.29 | 4.642 | 6.169 | 1.543 | 0.2026 | 0.2445 | 0.2470 | 0.8810 | 1.065 |
| Pixverse-V5.5 | ✗ | 11.54 | 4.558 | 5.982 | 1.824 | 0.1816 | 0.2305 | 0.2431 | 0.6627 | 1.306 |
| Ovi-1.1 | ✓ | 9.336 | 4.368 | 6.531 | 1.592 | 0.1620 | 0.1756 | 0.2391 | 0.9624 | 1.191 |
| JavisDiT | ✓ | 6.850 | 3.575 | 4.752 | – | 0.1284 | 0.1257 | 0.2320 | 1.322 | – |
| **- T2V + TV2A** | | | | | | | | | | |
| HunyuanVideo1.5 + Hunyuan-Foley | ✓ | 11.34 | 4.804 | 6.330 | – | 0.2598 | 0.2021 | 0.2436 | 0.8924 | – |
| Wan-2.2 + Hunyuan-Foley | ✓ | 13.43 | 5.605 | 6.353 | – | 0.2575 | 0.2076 | 0.2455 | 0.7935 | – |
| Wan-2.2 + MMAudio | ✓ | 13.43 | 5.605 | 6.076 | – | 0.2195 | 0.2448 | 0.2455 | 0.8890 | – |
| HunyuanVideo1.5 + MMAudio | ✓ | 11.34 | 4.804 | 6.101 | – | 0.2210 | 0.2466 | 0.2436 | 0.9427 | – |
| **- T2A + TA2V** | | | | | | | | | | |
| AudioLDM2 + MTV | ✓ | 8.066 | 3.458 | 6.253 | 1.264 | 0.1639 | 0.2698 | 0.2394 | 1.1592 | 0.6835 |

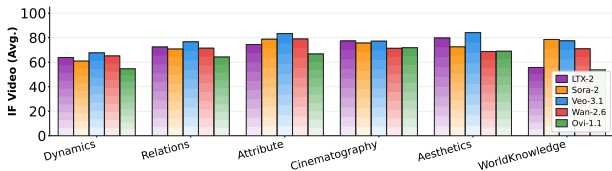

*Figure 6.* Sub-dimension comparison of IF Video.

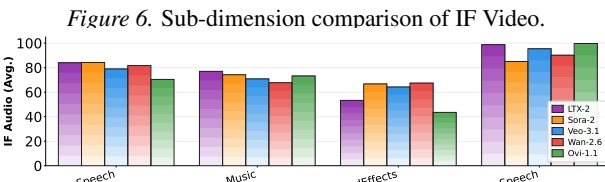

*Figure 7.* Sub-dimension comparison of IF Audio.

prone category, while Speech/Music-related requirements are comparatively easier to satisfy. This gap suggests that current systems struggle more with grounding diverse physical sound events to the prompt (and visual events) than with producing generic speech/music patterns.

**The Audio–Visual Realism Disconnect.** Figure 8 reveals a stark performance gap between modalities: while models demonstrate advanced visual integrity and temporal stability, audio realism remains a universal bottleneck, with artifacts, muffled timbres, and weak material–timbre grounding. Our follow-up failure decomposition suggests that this bottleneck is tied to fine-grained semantic content matching and temporal control as well as physical grounding; we therefore phrase this as diagnostic evidence rather than a definitive causal attribution (Appendix D.1).

**Difficulty Analysis** We analyze failure rates as prompt complexity increases along two axes: **Visual Subject Count** (Single/Multiple/Crowd; 1 / 2–3 / 4+ subjects) and **Event Temporal Structure** (Single/Sequential/Long-narrative; 1 / 2–3 / 4+ event units). As shown in Figure 9, failure rates increase monotonically with complexity. For subject count, failures rise from 8–10% (Single) to 21–44% (Crowd). Temporal structure exhibits a sharper escalation: Wan-2.6 increases from 22% failure (single event) to 63% failure for long-narrative prompts (4+ events), while LTX-2 reaches 80% failure. Overall, the 63–80% failure rates on long-narrative prompts across models identify long-horizon generation as a critical unsolved challenge. Failure also grows rapidly with event count, indicating a pronounced breakdown in temporal coherence beyond roughly 4–5 events.

**Human–MLLM Judge Agreement Analysis.** We evaluate automated-judging reliability on a 50-prompt subset by comparing MLLM scores with human ratings. For each prompt, both humans and the MLLM assign 1–5 scores on four dimensions: **IF Video**, **IF Audio**, **Video Realism**, and **Audio Realism**. We quantify judge–human disagreement using an **L1 distance** between the corresponding 4D score vectors (lower is better). As shown in Table 4, Gemini 2.5 Pro aligns most closely with human ratings on three dimensions (IF Video, IF Audio, and video Realism), while **audio realism** remains harder and therefore benefits from human verification. Full agreement protocol is in Appendix D.4.

**Validation via Human Pairwise Comparison.** Beyond absolute scores, we further validate whether the judge pre-

*Table 3.* Subjective evaluation performance over four dimensions. "IF" denotes instruction following.

| Method | Open-Source | IF Video↑ | IF Audio↑ | Video Realism↑ | Audio Realism↑ | Average↑ |
|---|---|---|---|---|---|---|
| **- T2AV** | | | | | | |
| Veo-3.1 | ✗ | 76.15 | 67.90 | 87.14 | 49.95 | 70.29 |
| Sora-2 | ✗ | 74.93 | 72.86 | 85.53 | 46.01 | 69.83 |
| Kling-2.6 | ✗ | 73.72 | 63.89 | 87.98 | 47.03 | 68.16 |
| Wan-2.6 | ✗ | 78.52 | 74.95 | 82.05 | 35.18 | 67.68 |
| Seedance-1.5 | ✗ | 60.96 | 61.22 | 88.94 | 53.84 | 66.24 |
| LTX-2 | ✓ | 63.97 | 64.74 | 89.95 | 36.23 | 63.72 |
| Wan-2.5 | ✗ | 76.56 | 57.95 | 76.00 | 35.06 | 61.39 |
| Pixverse-V5.5 | ✗ | 65.13 | 53.31 | 69.37 | 33.58 | 55.35 |
| Ovi-1.1 | ✓ | 55.05 | 52.83 | 65.93 | 30.75 | 51.14 |
| JavisDiT | ✓ | 32.56 | 15.26 | 34.97 | 14.85 | 24.41 |
| **- T2V + TV2A** | | | | | | |
| HunyuanVideo1.5 + Hunyuan-Foley | ✓ | 66.23 | 40.09 | 86.75 | 41.65 | 58.68 |
| Wan-2.2 + Hunyuan-Foley | ✓ | 64.54 | 37.10 | 89.63 | 41.25 | 58.13 |
| Wan-2.2 + MMAudio | ✓ | 64.79 | 38.19 | 89.63 | 36.05 | 57.17 |
| HunyuanVideo1.5 + MMAudio | ✓ | 66.10 | 35.94 | 85.38 | 35.15 | 55.64 |
| **- T2A + TA2V** | | | | | | |
| AudioLDM2 + MTV | ✓ | 47.13 | 54.39 | 56.73 | 31.90 | 47.54 |

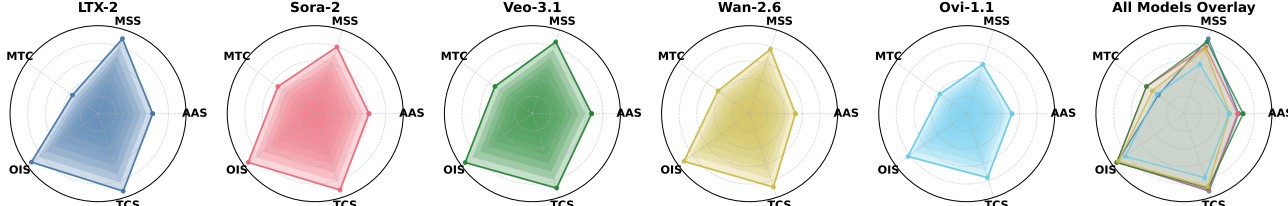

*Figure 8.* **Multi-metric radar comparison** on five complementary metrics on video and audio realism (higher is better).

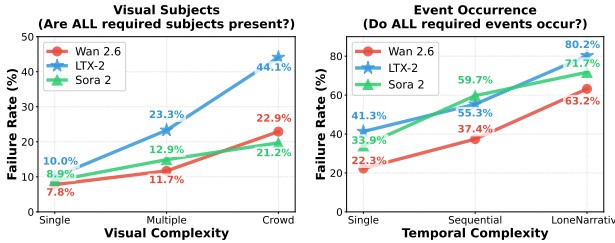

*Figure 9.* **Difficulty trends.** Failure rates increase with prompt complexity along Visual Subject Count and Event Count. The lower point indicates less failure in generated videos.

serves the **relative ordering** of models. We run forced-choice pairwise comparisons on the same 50 prompts across five T2AV systems (Veo-3.1, Wan-2.6, PixVerse-V5.5, Ovi-1.1, and JavisDiT). We fit a Bradley–Terry model to aggregate pairwise preferences into an Elo-style ranking. Gemini 2.5 Pro matches the human-derived ordering, indicating that it captures comparative judgments used for leaderboard construction. See Appendix D.5 for the Bradley–Terry fitting and the full head-to-head matrix.

**MLLM-as-a-Judge Robustness Analysis.** To assess the stability of our judge pipeline, we repeat the entire judging process three times on the same 500 video samples. For

*Table 4.* **L1 distance** between evaluators across four subjective dimensions (lower is better). The calculation details are in Equation 1 and Equation 2 in Appendix D.4.1. More detailed results are in Table 6(Appendix D.4.2).

| Evaluator | IF Video↓ | IF Audio↓ | Video Real.↓ | Audio Real.↓ | Overall↓ |
|---|---|---|---|---|---|
| Inter-Human | 0.917 | 0.911 | 0.926 | 1.042 | 0.949 |
| Gemini-2.5-Pro | **1.012** | **0.980** | **0.937** | 1.420 | **1.087** |
| Gemini-2.5-Flash | 1.212 | 1.027 | 1.397 | **1.193** | 1.207 |
| Qwen3-Omni-Flash | 1.026 | 1.887 | 1.297 | 1.680 | 1.473 |

each dimension, we compute the coefficient of variation (CV $= \sigma/\mu \times 100\%$) across the three runs, where a lower CV indicates more consistent scores. Gemini 2.5 Pro exhibits consistently low variability (CV $\leq 1.02\%$) across subjective dimensions, and Gemini 2.5 Flash yields a highly consistent five-system ranking (Spearman $\rho = 0.9248$), suggesting that the main comparative findings are not tied to one exact Gemini variant.

**Additional Analyses in Appendix.** We provide further analyses in appendix, including metric redundancy (Appendix D.3), human validation of AV synchronization ( Appendix D.1), preprocessing sensitivity (Appendix D.1), and an empirical upper-bound calibration using real-world videos (Appendix D.6).

## 5. Conclusion

We introduced T2AV-Compass, a unified benchmark for systematically evaluating text-to-audio-video generation. By combining a taxonomy-driven prompt construction pipeline with a dual-level evaluation framework, T2AV-Compass enables fine-grained and diagnostic assessment of video quality, audio quality, cross-modal alignment, instruction following, and realism. Extensive experiments demonstrate that our benchmark effectively differentiates model capabilities and exposes diverse failure modes in real-world settings.

## 6. Acknowledge

This work was supported by the Beijing Major Science and Technology Project (No. Z251100008425023). This work was supported in part by the National Natural Science Foundation of China (No. 62506161).

## Impact Statement

This paper introduces T2AV-Compass, a unified benchmark for Text-to-Audio-Video generation that reduces evaluation fragmentation and surfaces key capability gaps, including the Dynamics bottleneck and weak material–timbre consistency. By providing fine-grained diagnostics, our benchmark supports the development of more physically grounded, temporally coherent, and controllable audiovisual systems. While high-fidelity generation can amplify misinformation risks, standardized evaluation and diagnostic signals can also aid robustness auditing and complement forgery-detection research. Overall, T2AV-Compass advances more transparent and rigorous assessment for safer deployment of multimodal generative models.

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

## A. Limitations

Despite its comprehensiveness, T2AV-Compass is primarily constrained by the computational overhead of the MLLM-as-a-Judge protocol, which poses challenges for large-scale, real-time evaluation. Additionally, while our reasoning-first mechanism enhances interpretability, the evaluation remains subject to the intrinsic biases of the underlying MLLMs, such as preferences for specific visual styles or audio frequencies. Our judge protocol also depends on closed-source Gemini variants, so results may be affected by vendor updates, version drift, and model-specific preferences; we therefore report human agreement and Gemini Flash robustness checks, and treat the judge as a scalable proxy rather than a bias-free oracle. Lastly, the current prompt scale, while taxonomically diverse, may not fully capture the extreme long-tail distribution of rare physical interactions or niche artistic concepts.

## B. Future Work and Insight

The observed "Audio Realism Bottleneck" suggests that future research should improve temporal audio control, material-aware sound grounding, and native audiovisual modeling rather than relying only on composed pipelines. We plan to extend T2AV-Compass to long-duration videos ($> 10$ seconds), add explicit event-level sound-role attribution, and develop distilled lightweight evaluators that reduce the cost of repeated model-development evaluation.

# C. More Detailed Results

*Table 5.* Fine-grained comparison across sub-dimensions. A dash represent the model can't generate speech. (Part I)

| Method | Open | Attribute | | Cinematography | | | Sound | | | |
|---|---|---|---|---|---|---|---|---|---|---|
| | | Look↑ | Quantity↑ | Light↑ | Frame↑ | ColorGrading↑ | SFX↑ | Speech↑ | Music↑ | NonSpeech↑ |
| *- T2AV (end-to-end)* | | | | | | | | | | |
| Veo-3.1 | ✗ | 89.75 | 75.00 | 78.70 | 72.82 | 85.00 | 57.98 | 79.39 | 70.96 | 95.37 |
| Sora-2 | ✗ | 85.97 | 79.90 | 85.91 | 73.69 | 77.04 | 63.73 | 88.02 | 81.35 | 85.48 |
| Kling-2.6 | ✗ | 82.68 | 77.10 | 89.83 | 76.73 | 88.64 | 59.02 | 80.73 | 54.66 | 97.27 |
| Wan-2.6 | ✗ | 93.17 | 77.86 | 83.33 | 78.63 | 90.34 | 68.01 | 89.57 | 72.31 | 94.96 |
| SeeDance-1.5 | ✗ | 61.43 | 59.67 | 56.47 | 61.87 | 68.98 | 64.80 | 60.20 | 69.76 | 88.89 |
| LTX-2 | ✓ | 73.49 | 59.58 | 77.54 | 67.45 | 74.55 | 41.67 | 80.03 | 71.21 | 98.34 |
| Wan-2.5 | ✗ | 89.24 | 76.98 | 79.77 | 73.02 | 83.16 | 66.29 | 71.62 | 62.04 | 73.91 |
| PixVerse-V5.5 | ✗ | 73.75 | 54.34 | 74.49 | 68.31 | 63.89 | 48.16 | 80.50 | 54.07 | 77.19 |
| Ovi-1.1 | ✓ | 62.65 | 51.64 | 72.03 | 56.81 | 78.64 | 29.34 | 62.99 | 66.55 | 99.61 |
| JavisDiT | ✓ | 36.36 | 39.62 | 42.95 | 46.13 | 49.07 | 29.87 | – | 15.91 | – |
| *- T2V + TV2A* | | | | | | | | | | |
| HunyuanVideo1.5 + Hunyuan-Foley | ✓ | 80.72 | 66.12 | 70.13 | 73.02 | 80.45 | 47.87 | – | 72.41 | – |
| Wan-2.2+MMAudio | ✓ | 76.36 | 62.85 | 76.69 | 71.16 | 76.36 | 46.63 | – | 64.66 | – |
| Wan-2.2 + Hunyuan-Foley | ✓ | 76.36 | 65.65 | 80.30 | 71.41 | 71.82 | 48.35 | – | 66.21 | – |
| HunyuanVideo1.5 + MMAudio | ✓ | 81.33 | 67.29 | 72.46 | 75.00 | 82.27 | 44.90 | – | 62.93 | – |
| *- T2A + TA2V* | | | | | | | | | | |
| AudioLDM2 + MTV | ✓ | 65.81 | 55.37 | 58.26 | 61.01 | 57.27 | 39.88 | 48.18 | 77.07 | 98.05 |

*Table 6.* Fine-grained comparison across sub-dimensions. A dash represent the model can't generate speech.(Part II)

| Method | Open | Dynamics | | | | Relations | | Aesthetics | | World Knowledge |
|---|---|---|---|---|---|---|---|---|---|---|
| | | Camera Motion↑ | Inter-action↑ | Motion↑ | Trans-form↑ | Spatial↑ | Logical↑ | Style↑ | Mood↑ | Factual↑ |
| *- T2AV (end-to-end)* | | | | | | | | | | |
| Veo-3.1 | ✗ | 52.50 | 68.72 | 67.48 | 54.20 | 72.15 | 78.52 | 81.34 | 87.50 | 75.18 |
| Sora-2 | ✗ | 43.65 | 65.62 | 60.80 | 54.69 | 76.47 | 67.41 | 74.09 | 84.59 | 80.29 |
| Kling-2.6 | ✗ | 63.96 | 67.21 | 61.76 | 57.65 | 73.63 | 71.29 | 77.83 | 85.67 | 60.52 |
| Wan-2.6 | ✗ | 75.62 | 65.50 | 61.79 | 57.66 | 76.71 | 75.18 | 74.24 | 87.13 | 79.72 |
| SeeDance-1.5 | ✗ | 67.62 | 66.08 | 59.12 | 55.64 | 53.12 | 53.08 | 74.66 | 81.88 | 49.30 |
| LTX-2 | ✓ | 68.87 | 47.34 | 51.85 | 46.08 | 70.70 | 61.13 | 67.50 | 87.50 | 44.66 |
| Wan-2.5 | ✗ | 74.32 | 67.69 | 60.02 | 57.12 | 77.85 | 77.76 | 72.55 | 86.00 | 75.74 |
| PixVerse-V5.5 | ✗ | 59.16 | 71.18 | 64.52 | 46.88 | 69.74 | 64.89 | 65.52 | 81.52 | 56.54 |
| Ovi-1.1 | ✓ | 53.11 | 40.68 | 39.44 | 35.82 | 60.55 | 50.81 | 54.17 | 73.78 | 42.07 |
| JavisDiT | ✓ | 12.45 | 18.60 | 20.34 | 15.04 | 25.00 | 26.97 | 27.35 | 42.19 | 33.92 |
| *- T2V + TV2A* | | | | | | | | | | |
| HunyuanVideo1.5+Hunyuan-Foley | ✓ | 71.51 | 52.15 | 53.15 | 41.98 | 70.90 | 61.13 | 70.67 | 72.26 | 57.24 |
| Wan-2.2 + Hunyuan-Foley | ✓ | 56.79 | 50.10 | 46.48 | 38.99 | 72.85 | 61.45 | 67.17 | 82.01 | 51.90 |
| Wan-2.2+MMAudio | ✓ | 57.17 | 48.05 | 49.63 | 38.06 | 70.70 | 63.23 | 66.67 | 82.62 | 52.76 |
| HunyuanVideo + MMAudio | ✓ | 69.06 | 46.62 | 50.93 | 40.49 | 68.36 | 64.03 | 72.83 | 66.16 | 57.59 |
| *- T2A + TA2V* | | | | | | | | | | |
| AudioLDM2 + MTV | ✓ | 27.92 | 29.41 | 32.59 | 29.85 | 47.66 | 47.74 | 40.33 | 55.18 | 37.93 |

# D. More Analysis Experiments

## D.1. Validation and Diagnostic Additions

**Audiovisual synergy checklist coverage.**    Among 970 audio-related checklist questions, 307 (31.65%) require audiovisual synergy, such as verifying whether sound sources, timing, or material cues match visible events rather than audio-only presence. This coverage complements the objective cross-modal metrics by explicitly testing joint audiovisual interaction in the checklist-based diagnosis.

**Human validation of AV synchronization.**    For objective cross-modal validation, five annotators rated audio-video synchronization on a held-out subset; pairwise human Spearman correlation is 0.8660, and the corresponding synchronization metric reaches SRCC = 0.9172 against mean human scores.

**Preprocessing and configuration sensitivity.**    We tested objective-metric sensitivity on 100 videos. Resampling audio from 48 kHz to 16 kHz changes T–A (CLAP), A–V (ImageBind), PQ, and SQ by only 0.0004, 0.0005, 0.0015, and 0.0015, respectively. For video, DOVER VT scores at 24, 12, and 8 fps are 0.7543, 0.7519, and 0.7551. These controlled results suggest that moderate native sampling-rate or frame-rate differences do not confound our main objective-alignment conclusions, while the released evaluator will standardize preprocessing (e.g., audio resampling to 48 kHz) for rigorous comparison.

**Audio failure decomposition.**    A follow-up decomposition suggests that audio failures reflect fine-grained semantic content matching and temporal control in addition to physical grounding. Biological sounds are hardest (50.3% failure), musical sounds easiest (40.5%), and mechanical sounds near average (42.9%). Sequential prompts are harder than simultaneous prompts (48.8% vs. 40.7%), even though simultaneous mixtures can be acoustically complex. Off-screen prompts are often easier because they are dominated by music and contain fewer required audio elements.

## D.2. Detailed Analysis by Model Capability

Based on the objective metrics reported in Table 2, we provide a concise summary of the performance characteristics for each evaluated system:

**Proprietary End-to-End Models**

- **Veo-3.1:** Demonstrates high performance across most dimensions, particularly achieving a high Video Technical score (VT 13.39) while maintaining consistent audio-visual alignment.

- **Seedance-1.5:** Achieves the highest scores in Audio Aesthetics (AA 7.403), A-V Alignment (0.2875), and Lip Sync (1.560), indicating strong performance in cross-modal coherence.

- **PixVerse-V5.5:** Exhibits the best performance in Temporal Synchronization (DS 0.6627) and Speech Quality (SQ 1.824), despite recording average scores in visual fidelity.

- **Wan-2.5 / Wan-2.6:** These models show strong visual fidelity (VT) and text-video alignment, while recording comparatively lower scores in audio quality and synchronization metrics.

- **Sora-2:** Records lower scores on objective signal-level metrics (VT, AA), contrasting with its performance in semantic instruction following.

- **Kling-2.6:** Maintains a balanced profile across video and audio metrics, with relatively high speech quality (SQ 1.783).

**Open-Source End-to-End Models**

- **LTX-2:** Shows lower low-level visual fidelity (VT 7.16) while maintaining competitive audio generation capabilities compared to other open-source models.

- **Ovi-1.1:** Displays consistent performance across all evaluated dimensions without significant variance between modalities.

- **JavisDiT:** Scores lower in visual quality metrics and does not support speech generation.

**Composed Generation Pipelines**

- **Wan-2.2 + Audio Modules:** This pipeline achieves the highest Visual Technical scores (VT 13.43) due to the T2V backbone, but exhibits high temporal synchronization error (DS $\approx$ 0.89).

- **AudioLDM2 + MTV:** Achieves the highest Text-Audio alignment (0.2698) but records lower scores in video quality metrics.

### D.3. Metric Independence and Redundancy

To justify the necessity of the metrics in our multi-dimensional evaluation framework, we analyze the orthogonality among the proposed metrics. To prevent the model's "general capability factor" from obscuring the "specificity" of individual metrics, we apply a de-meaning procedure to the evaluation results. Specifically, for each video sample, we compute the Z-scores of all metrics and subtract the model's average metric score on that particular sample. This process removes the effect of overall model capability and thus highlights the trade-offs among metrics across different dimensions.

The Pearson correlation heatmap computed on the processed metrics is shown in Figure 10. The absolute correlation coefficients between most metric pairs remain below 0.3, indicating a high degree of orthogonality.

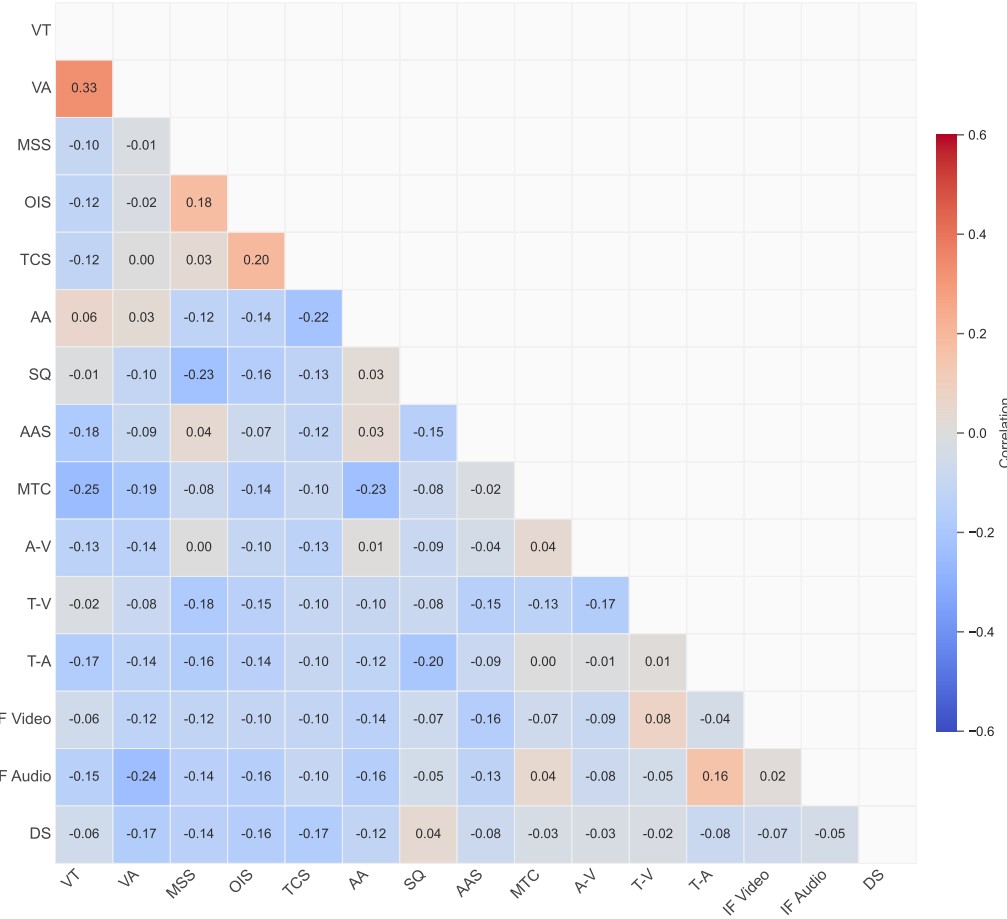

*Figure 10.* **Metric correlation heatmap.** We compute Pearson correlations on per-sample *de-meaned* z-scored metrics to factor out overall model strength. Most pairs show weak correlation ($\|r\| < 0.3$), suggesting limited redundancy among objective and subjective dimensions.

## D.4. Human-MLLM Agreement Analysis: Methodology and Extended Results

### D.4.1. EXPERIMENTAL SETUP

**Dataset**  We evaluated 50 diverse prompts from the Consistency50 benchmark across 5 state-of-the-art T2AV models: OVI-1.1, JavisDiT, Wan2.6, PixverseV5.5, and Veo3.1. Each prompt generated one video-audio pair per model, yielding 250 total samples.

**Human Annotation Protocol**  Five trained annotators independently rated each sample on a 5-point Likert scale (1=poor, 5=excellent) across four dimensions:

- **IF Video**: Visual adherence to prompt specifications (objects, actions, scene composition)

- **IF Audio**: Audio alignment with prompt requirements (sound sources, acoustic events)

- **Video Realism**: Motion stability, temporal consistency, absence of visual artifacts

- **Audio Realism**: Material-timbre consistency and environmental acoustic plausibility

Five annotators followed a detailed rubric with exemplar videos for each score level. Inter-annotator training sessions ensured consistent interpretation of criteria.

**MLLM-Judge Configuration**  We evaluated three MLLMs using identical prompting strategies:

For instruction-following dimensions, MLLMs answered structured checklist questions; realism dimensions used direct quality assessment prompts. All evaluations were conducted offline to eliminate network latency effects.

**Inter-Human Agreement**  For each dimension $d$, we computed pairwise L1 distances across all valid annotator pairs:

$$\text{L1}_{\text{H-H}}^{(d)} = \frac{1}{N_{\text{pairs}}^{(d)}} \sum_{(i,m)\in\mathcal{S}} \sum_{\substack{a_1,a_2\in\mathcal{A}_{i,m} \\ a_1<a_2}} \left| S_{a_1}^{(d)}(i,m) - S_{a_2}^{(d)}(i,m) \right| \tag{1}$$

where $\mathcal{S}$ is the sample set, $\mathcal{A}_{i,m}$ denotes annotators who rated sample $(i,m)$, $S_a^{(d)}$ is annotator $a$'s score, and $N_{\text{pairs}}^{(d)}$ is the total number of valid pairs (503 per dimension).

**Human-MLLM Agreement**  Human consensus $C_{\text{H}}$ was computed as the mean across annotators per sample. MLLM agreement was then:

$$\text{L1}_{\text{H-MLLM}}^{(d)} = \frac{1}{N^{(d)}} \sum_{(i,m)\in\mathcal{S}^{(d)}} \left| C_{\text{H}}^{(d)}(i,m) - S_{\text{MLLM}}^{(d)}(i,m) \right| \tag{2}$$

where $N^{(d)} = 500$ for instruction-following dimensions (all 5 models $\times$ 50 prompts $\times$ 2 checklist aggregations) and $N^{(d)} = 250$ for realism dimensions (single score per sample).

### D.4.2. COMPREHENSIVE RESULTS

**Primary Agreement Analysis**  Table 6 presents complete agreement statistics including standard deviations and sample sizes. Gemini 2.5 Pro achieves the lowest overall L1 distance (1.087), with near-human performance on Video Realism (0.937 $\pm$ 0.864)—only 1.1% worse than the inter-human baseline (0.926 $\pm$ 0.929). Notably, Qwen3-Omni exhibits catastrophic failure on IF Audio (L1=1.887, 107% worse than human baseline), indicating fundamental limitations in audio instruction verification.

**MLLM-MLLM Consistency**  Table 7 reveals that same-family MLLMs (Gemini variants) achieve **higher mutual agreement** (L1=0.702) than inter-human agreement (L1=0.949), suggesting shared architecture/training induces consistent evaluation standards. Cross-family pairs show substantially higher disagreement (L1=1.322–1.476), indicating divergent implicit quality criteria across model families.

*Table 6.* Complete agreement analysis. Lower L1 indicates better agreement. Bold values denote best MLLM performance per dimension.

| Evaluator | Dimension | Mean L1 | Std | Gap | Gap % |
|---|---|---|---|---|---|
| Inter-Human | IF Video | 0.917 | 0.994 | – | – |
| | IF Audio | 0.911 | 0.998 | – | – |
| | Video Realism | 0.926 | 0.929 | – | – |
| | Audio Realism | 1.042 | 0.899 | – | – |
| Gemini-2.5-Pro | IF Video | 1.012 | 0.942 | +0.095 | +10.4% |
| | IF Audio | **0.980** | 0.876 | +0.069 | +7.6% |
| | Video Realism | **0.937** | 0.864 | +0.011 | +1.1% |
| | Audio Realism | 1.420 | 1.102 | +0.378 | +36.3% |
| | **Overall** | **1.087** | 0.947 | +0.138 | +14.5% |
| Gemini-2.5-Flash | IF Video | 1.212 | 1.087 | +0.295 | +32.2% |
| | IF Audio | 1.027 | 0.912 | +0.116 | +12.7% |
| | Video Realism | 1.397 | 1.023 | +0.471 | +50.9% |
| | Audio Realism | **1.193** | 0.954 | +0.151 | +14.5% |
| | Overall | 1.207 | 1.019 | +0.258 | +27.2% |
| Qwen3-Omni | IF Video | 1.026 | 1.003 | +0.109 | +11.9% |
| | IF Audio | 1.887 | 1.321 | +0.976 | +107.1% |
| | Video Realism | 1.297 | 1.087 | +0.371 | +40.1% |
| | Audio Realism | 1.680 | 1.203 | +0.638 | +61.2% |
| | Overall | 1.473 | 1.154 | +0.524 | +55.2% |

*Table 7.* Pairwise MLLM-MLLM agreement on instruction-following dimensions (N=500 per pair).

| MLLM Pair | Mean L1 | Std Dev | Comparison Type |
|---|---|---|---|
| Gemini-2.5-Flash vs Gemini-2.5-Pro | **0.702** | 1.278 | Same-family |
| Gemini-2.5-Pro vs Qwen3-Omni | 1.322 | 1.662 | Cross-family |
| Gemini-2.5-Flash vs Qwen3-Omni | 1.476 | 1.739 | Cross-family |

## D.5. Arena50 Win Rate Analysis

### D.5.1. EXPERIMENTAL DESIGN

**Dataset**  The Arena50 benchmark comprises 50 prompts spanning object generation, motion dynamics, multi-character interactions, and audio-visual synchronization scenarios. Each prompt was rendered by five T2AV models: Veo3.1, Wan2.6, PixVerseV5.5, OVI1.1, and Javisdit, yielding 250 total video-audio pairs.

**Human Evaluation Protocol**  Five trained annotators performed forced-choice pairwise comparisons under identical prompts. For each prompt, annotators evaluated four randomly sampled model pairs across two subjective dimensions:

- **Instruction Following**: Which video better adheres to the textual prompt specifications?

- **Realism**: Which video exhibits superior motion stability, temporal coherence, and audio-visual plausibility?

### D.5.2. RANKING METHODOLOGY

**Bradley–Terry Model**  We model pairwise win probabilities using the Bradley–Terry (BT) framework (Bradley & Terry, 1952). For models $i$ and $j$ with latent strengths $\theta_i, \theta_j$, the probability that $i$ beats $j$ is:

$$P(i \succ j) = \frac{e^{\theta_i}}{e^{\theta_i} + e^{\theta_j}} \tag{3}$$

Maximum likelihood estimation (MLE) solves for $\theta$ given observed win counts $w_{ij}$:

$$\hat{\theta} = \arg\max_{\theta} \sum_{i \neq j} w_{ij} \log \left( \frac{e^{\theta_i}}{e^{\theta_i} + e^{\theta_j}} \right) \tag{4}$$

**Elo Score Conversion**   BT parameters $\theta$ were converted to standard Elo scores with OVI1.1 as baseline (Elo = 1500):

$$\text{Elo}_i = 1500 + 400 \times (\theta_i - \theta_{\text{OVI1.1}}) \tag{5}$$

This enables intuitive interpretation: a 400-point Elo gap implies a 90.9% predicted win probability.

### D.5.3. COMPLETE RESULTS

**Human-Derived Rankings.**   Table 8 reports the model ranking derived from human *pairwise* preferences on the Arena50 subset. For each prompt, annotators perform forced-choice comparisons between model outputs; we aggregate all pairwise outcomes and fit a Bradley–Terry (BT) model to estimate a latent *skill* parameter for each model. We then convert these BT skill parameters into Elo-style scores (with a fixed reference point) for interpretability; higher Elo and win rate indicate stronger human preference. Veo-3.1 ranks first with an Elo of 2607.2 (win rate: 89.45%), followed by Wan-2.6 (2202.6, 72.54%); PixVerse-V5.5 is mid-tier (1804.9, 52.24%), while Ovi-1.1 (1500.0) and JavisDiT (209.2) form the lower tier. The 2,398-point gap between the top and bottom models suggests substantial quality variation under human judgments.

*Table 8.* Human-derived Elo rankings from Arena50 pairwise comparisons. Win rates computed against all opponents.

| Rank | Model | Elo | Win Rate | BT Param |
|------|-------|-----|----------|----------|
| 1 | Veo-3.1 | 2607.2 | 89.45% | 2.768 |
| 2 | Wan-2.6 | 2202.6 | 72.54% | 1.757 |
| 3 | PixVerse-V5.5 | 1804.9 | 52.24% | 0.762 |
| 4 | Ovi-1.1 | 1500.0 | 38.92% | 0.000 |
| 5 | JavisDiT | 209.2 | 1.63% | -3.227 |

**Pairwise Comparison Matrix**   Table 9 shows head-to-head win rates. Veo3.1 dominates all opponents (minimum 76.5% vs Wan2.6), while Javisdit loses consistently ($< 2\%$ win rate against all models). The PixVerseV5.5 vs OVI-1.1 matchup (69.0% for PixVerseV5.5) represents the most competitive pairing.

*Table 9.* Head-to-head win rates (%) from human pairwise comparisons. Rows = winner, columns = loser. Diagonal omitted.

| Winner \ Loser | Veo3.1 | Wan2.6 | PixVerseV5.5 | OVI1.1 | Javisdit |
|------|--------|--------|--------------|--------|----------|
| Veo-3.1 | – | 76.5 | 85.4 | 95.6 | 98.4 |
| Wan-2.6 | 23.5 | – | 77.3 | 85.4 | 98.4 |
| PixVerse-V5.5 | 14.6 | 22.7 | – | 69.0 | 97.9 |
| Ovi-1.1 | 4.4 | 14.6 | 31.0 | – | 94.4 |
| JavisDiT | 1.6 | 1.6 | 2.1 | 5.6 | – |

### D.6. Upper Bound Analysis of MLLM-as-a-Judge

To validate the discriminative power of our checklist-based evaluation framework and establish an empirical performance ceiling, we conducted a calibration test using 100 high-quality real-world video clips. By treating these "ground-truth" videos as inputs, we assessed the MLLM judge's capability to recognize idealized **IF Video** and **IF Audio** performance in Figure 11. This analysis yields several key insights into the reliability of our benchmark.

First, the significant score margin between real-world videos and SOTA models—particularly in *Relations* (97.44 vs 77.81) and *Dynamics* (90.41 vs 65.14)—demonstrates the **high sensitivity** of our checklist items. It proves that the MLLM judge is not prone to "over-praising" generated content and can effectively identify the gap in complex interaction logic and temporal motion consistency.

Second, the high absolute scores achieved by real-world videos (exceeding 90 in most IFV dimensions) serve as a **sanity check** for the evaluation metrics; it confirms that the requirements defined in our checklists are realistic and attainable, rather than being grounded in hallucinatory standards. However, the relatively lower ceiling in *Sound Effects* (83.50) suggests a potential "bottleneck" in the judge's auditory perception, where the MLLM may be overly critical of subtle environmental noise even in authentic recordings.

Finally, the proximity of scores in *Aesthetics* (84.42 vs 84.83) reveals that while SOTA models have nearly mastered individual frame quality, the core challenge remains the **structural and logical alignment** reflected in the larger gaps of

other dimensions. This calibration ensures that our framework provides a reliable and scalable roadmap for future generative AI development.

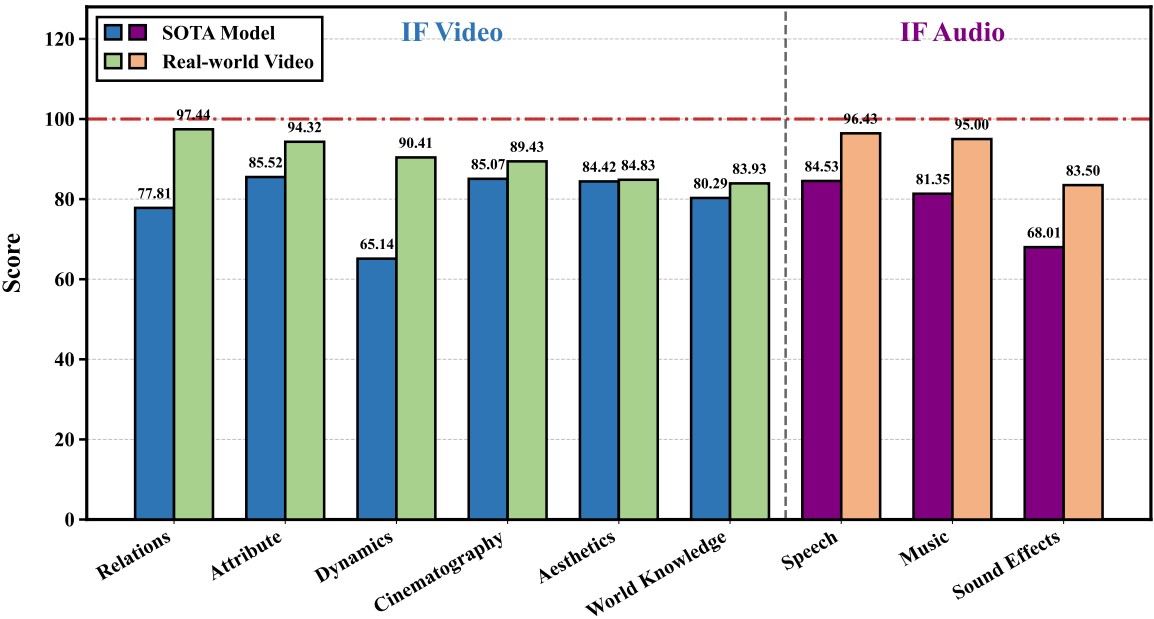

*Figure 11.* Sota VS Real

# E. MLLM Evaluation Framework

This section details the methodology for evaluating models based on the ability to follow instructions. The evaluation includes both visual and audio content instruction following, along with specific metrics for each modality.

## E.1. Instruction Following Metrics Calculation

This subsection covers the calculation methodology for the Instruction Following metrics: IF-Video and IF-Audio.

### E.1.1. SCORE NORMALIZATION

All raw evaluation scores are collected on a 5-point Likert scale (1-5) and normalized to the range [0, 1] using the formula:

$$S_{\text{norm}} = \frac{S_{\text{raw}} - 1}{4} \tag{6}$$

where $S_{\text{raw}}$ is the original score and $S_{\text{norm}}$ is the normalized score.

### E.1.2. IF-VIDEO CALCULATION

IF-Video measures the model's ability to follow visual content instructions. It is computed as the arithmetic mean of all non-audio major category scores:

$$\text{IF-Video} = \frac{1}{N} \sum_{i=1}^{N} C_i \tag{7}$$

where $N$ is the number of visual categories (e.g., Object, Attribute, Motion, Scene, etc.), and $C_i$ represents the average normalized score for category $i$. Each category score $C_i$ is itself the mean of its sub-dimension scores:

$$C_i = \frac{1}{M_i} \sum_{j=1}^{M_i} S_{ij} \tag{8}$$

where $M_i$ is the number of sub-dimensions in category $i$, and $S_{ij}$ is the normalized score for the $j$-th sub-dimension.

### E.1.3. IF-AUDIO CALCULATION

IF-Audio evaluates the model's adherence to audio-related instructions with a penalty mechanism for unwanted speech. The formula incorporates four audio sub-dimensions:

$$\text{IF-Audio} = \frac{(S_{\text{speech}} - P) + S_{\text{sfx}} + S_{\text{music}}}{3} \tag{9}$$

where:

- $S_{\text{speech}}$: normalized score for Speech quality

- $S_{\text{sfx}}$: normalized score for Sound Effects quality

- $S_{\text{music}}$: normalized score for Music quality

- $P$: penalty term computed as $P = 1 - S_{\text{nonspeech}}$

- $S_{\text{nonspeech}}$: normalized score for NonSpeech compliance (default 1.0 if not evaluated)

The penalty term $P$ penalizes the speech score when speech appears in contexts where it should not be present.

### E.1.4. FINAL SCORE REPORTING

Both metrics are reported in the range [0, 1] and can be scaled to [0, 100] for presentation:

$$\text{Score}_{\text{final}} = \text{Score}_{\text{normalized}} \times 100 \tag{10}$$

### E.2. Instruction Following Dimensions

### E.3. Realism

The framework for realism assessment is divided into five key dimensions: TCS (Temporal Coherence Score), OIS (Object Integrity Score), MTC (Material–Timbre Consistency), MSS (Motion Smoothness Score), and AAS (Acoustic Artifact Score). Each dimension captures critical factors that ensure the realism of the generated content, such as temporal coherence, structural integrity, sound-to-material matching, and the absence of motion/audio artifacts.

## F. Experimental Setup

**Implementation Details**   For video generation, we adhere to the native configurations of each T2AV system to preserve their intended technical quality. Specifically, we utilize the default video frame rates and audio sampling rates provided by the original models. Regarding spatial resolution, most systems are configured to output at 720p. To ensure a comprehensive comparison, we set Javis to 480p resolution, while Kling-2.6 is leveraged at 1080p. Notably, for systems based on Ovi-1.1 and composed generation, we employ Gemini-2.5-Pro to paraphrase the original prompts. This ensures the input text aligns with the specific reasoning and instruction-following requirements of their respective generation pipelines, thereby facilitating a more equitable performance assessment.

**Evaluation Settings**   To ensure reproducibility and minimize variance in the subjective assessment, we configure the MLLM judge with a deterministic decoding strategy (do_sample=False, temperature=0). For visual processing, the judge samples frames at a default rate of **2 FPS**, striking a balance between capturing motion dynamics and managing token overhead. For objective synchronization analysis, we employ the *desync* tool with an default `offset_sec` of **-2.0** to define the temporal search window.

*Table 10.* IF Video dimensions

| Dimension | Sub-dimension | Definition |
|---|---|---|
| **World Knowledge** | Factual Knowledge | Accurate depiction of inherent characteristics of specific entities, landmarks, and historical/cultural symbols |
| **Attribute** | Look | Visual appearance including color, shape, size, material, expression, and physical state |
| | Quantity | Statistical count of specific objects in the frame |
| **Cinematography** | Light | Light sources, lighting types (backlighting, volumetric light), and light/shadow texture |
| | Frame | Shot size, lens settings, shooting angle, and framing methods |
| | Color Grading | Color tendency, saturation, and contrast style |
| **Dynamics** | Motion | Specific behaviors, speed, and trajectory characteristics of objects |
| | Interaction | Contact or reactions between multiple entities |
| | Transformation | Changes in essential attributes or state evolution over time |
| | Camera Motion | Movement methods of the camera (dolly, pan, track, etc.) |
| **Relations** | Spatial | Physical positional relationships in 2D plane and 3D depth |
| | Logical | Abstract semantic connections including composition, comparison, and inclusion |
| **Aesthetics** | Style | Visual expression form, artistic genre, or medium texture |
| | Mood | Overall emotional tone or environmental atmosphere |

# G. More Data distribution.

# H. Detailed Dataset Construction

## H.1. Crawl From Existing Data

To establish a foundation of broad semantic coverage, we aggregate raw prompts from a variety of high-quality, diverse sources, including VidProM, the Kling AI community, LMArena, and Shot2Story. This multi-source aggregation ensures that T2AV-Compass captures a wide spectrum of user intents, ranging from creative storytelling to rigorous physical interaction scenarios.

**Semantic Denoising and Diversity Enhancement**   To mitigate the inherent imbalance between common concepts (e.g., generic landscapes) and long-tail distributions (e.g., specific causal physical events), we implement a sophisticated semantic clustering and sampling strategy:

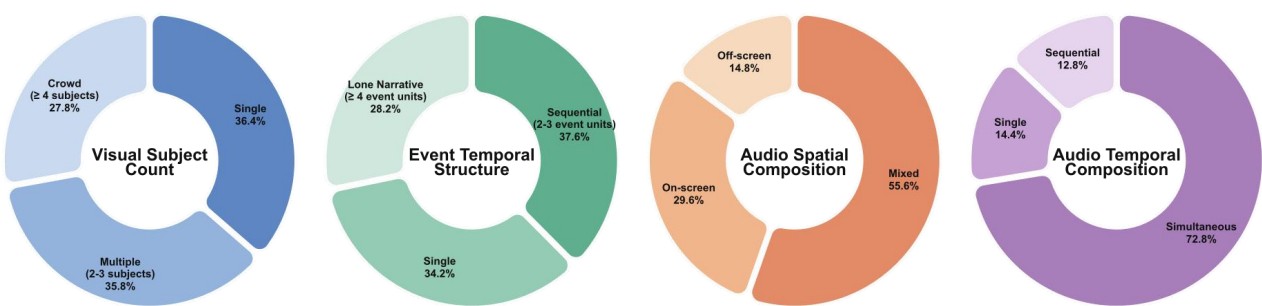

*Figure 12.* **Dataset statistics of T2AV-Compass.** Distributions of audiovisual complexity factors, including Visual Subject Count, Event Temporal Structure, Audio Spatial Composition, and Audio Temporal Composition.

*Table 11.* IF Audio dimensions

| Dimension | Sub-dimension | Definition |
| --- | --- | --- |
| **Sound** | Sound Effects | Ambient atmosphere sounds and specific physical sounds triggered by actions |
| | Speech | Human oral expression including dialogue, monologue, and voiceover |
| | NonSpeech | Human speech not required unless explicitly specified |
| | Music | Music-related elements including instruments, genres, rhythm, and emotion |

- **Embedding and Deduplication:** We encode all candidate prompts into a high-dimensional semantic space using all-mpnet-base-v2. To eliminate redundant entries while preserving subtle stylistic variations, we perform aggressive deduplication using a cosine similarity threshold of 0.8.

- **Square-root Sampling:** We cluster the deduplicated prompts and apply a square-root sampling strategy, where the sampling probability is defined as $P(c) \propto 1/\sqrt{|S_c|}$ ($|S_c|$ being the cluster size). This approach effectively suppresses over-represented topics and elevates the visibility of niche, semantically distinctive scenarios, ensuring the benchmark is not biased toward frequent but simplistic prompts.

## H.2. Real Video Collection

**Real-World Reference Collection**  To establish an empirical "realism ceiling" and provide a high-fidelity baseline for the *T2AV-Compass* benchmark, we curated a diverse collection of authentic videos sourced from **YouTube**. Each entry is meticulously documented with its source URL and precise temporal segments (start and end timestamps) to ensure full reproducibility. We implemented a multi-stage expert filtering pipeline based on the following criteria:

- **Spatiotemporal Specifications:** All videos strictly adhere to a 16:9 aspect ratio with a minimum native resolution of 720p (unified to 720p in post-processing). To capture meaningful semantic units, clips are trimmed to 5–10 seconds. The content spans various complexities, ranging from single-event visual atoms (e.g., a person smiling) to intricate, sequential narratives involving four or more logical steps (e.g., a person picking up a glass and dropping it).

- **Anti-Overfitting & Integrity:** We prioritized content published after October 2024 to mitigate potential data leakage from the training sets of current T2AV models. We manually excluded User-Generated Content (UGC) with watermarks, heavy text overlays, or rapid montage-style editing ($\leq 2$ cuts per clip), while retaining scene-inherent text necessary for narrative context.

- **Audiovisual Complexity:** The collection emphasizes rich, multi-layered soundscapes featuring 1–4 distinct sound effects and essential off-screen audio (e.g., ambient BGM or narration). Notably, 30% of the videos contain coherent human speech, while 70% incorporate diverse camera dynamics, including linear translation, angular rotation, zooming, and realistic handheld jitters, to test the models' ability to handle non-static viewpoints.

- **Thematic Distribution:** The reference set mirrors our proposed taxonomy across seven domains: *Modern Life & Drama* (25%), *Documentary & History* (23%), *Fantasy* (17%), *Sci-Fi* (14%), *Animation* (14%), *Horror & Humor* (14%), and *Commercial & Promotion* (13%).

*Table 12.* Technical Quality Evaluation Framework: Five Scoring Dimensions and Sub-dimensions

| Dimension | Sub-dimension | Definition |
|---|---|---|
| **TCS** | Existence Continuity | Detecting abnormal disappearance, appearance, and flickering of objects |
| | Identity Stability | Examining sudden changes in object category or appearance features |
| | Occlusion & Boundary Logic | Verifying logical consistency of object occlusion and frame entry/exit |
| **OIS** | Biological Anatomical Constraints | Consistency of limb length, joint angles, and facial features |
| | Rigid Body Rigidity | Geometric shape preservation and contour stability of rigid objects |
| | Texture & Semantic Consistency | Frame-to-frame consistency of texture details and surface patterns |
| **MTC** | Material-Timbre Matching | Accuracy of sound timbre in representing material properties |
| | Interaction Dynamics | Correspondence between sound envelope and action intensity |
| | Environmental Acoustics | Matching of reverb and echo with visual spatial characteristics |
| **MSS** | Artifacts & Degradation | Detection of unnatural blur, pixel blocks, and flickering |
| | Fluidity of Motion | Smoothness of frame transitions and optical flow consistency |
| | Scene-Aware Analysis | Context-appropriate evaluation of motion blur based on scene dynamics |
| **AAS** | Generative Artifacts | Detection of metallic sound, smearing, and frequency truncation |
| | Temporal Stability | Detection of pops, clicks, dropouts, and noise floor pumping |
| | Signal Integrity | Detection of clipping distortion and hallucinated noises |

# I. Prompts

In this section, we present all the prompts that were used throughout the process.

## I.1. Rewrite Prompt

---

**Role & Task**
*Instruction prompt for professional T2AV prompt rewriting*

```
**Your Role and Task:**
You are an expert in Text-to-Audio/Video model prompt optimization. Your task is to
↪  receive the original Text-to-Audio/Video prompt provided by the user, and strictly
↪  follow the detailed rewriting guidelines below to rewrite it into a structured,
↪  visually strong, and naturally fluent professional-level prompt. The rewritten
↪  prompt should be output directly without any pre or post explanatory text.

**Core Rewriting Principles:**

1.  **Absolute Fidelity to Original and Adaptive Enhancement**: Your primary principle
↪  is to maximize the retention of the original prompt's core creativity, theme, and
↪  emotion. Your work is to "enhance" rather than "replace". Please dynamically decide
↪  which core description modules need to be selected based on the core concept of the
↪  original Prompt, and make reasonable and creative expansions.
2.  **Distinguish Prompt from API Parameters**: You must understand that video
↪  **resolution (size)**, **aspect ratio**, and **duration (seconds)** are directly
↪  controlled by API call parameters. Therefore, in the rewritten prompt, **strictly
↪  prohibit** any descriptions that specify or imply these parameters. Your prompt
↪  must focus on all other factors: **subject, scene, dynamics, cinematography, style,
↪  and sound**.

**Detailed Rewriting Guidelines and Methodology:**

You will analyze the user's original prompt and **selectively integrate** the following
↪  six core modules based on specific situations to construct a natural, fluent, and
↪  logically coherent paragraph.

1.  **Subject (Subject Description) [Core Element]:**
    *   **Goal:** Clearly depict the video focus.
    *   **Method:** Use descriptive phrases, not just nouns, to define its
    ↪  **appearance, clothing, features, and essence**. For example, refine "a girl"
    ↪  to "**A black-haired Miao girl wearing intricate ethnic minority clothing**";
    ↪  refine "a monster" to "**A flying fairy from another world, dressed in tattered
    ↪  yet elegant attire, with a pair of strange wings made of rubble fragments**".

2.  **Scene (Scene Description) [Core Element]:**
    *   **Goal:** Build an environment with depth and atmosphere.
    *   **Method:** Specifically describe **environment, background, foreground, and
    ↪  light atmosphere**. For example, "**A sunlit Roman square with actors seated
    ↪  around a marble table, a horse-drawn carriage passing on the cobblestone street
    ↪  in the background**".

3.  **Motion (Dynamic Description) [Core Element]:**
    *   **Goal:** Inject vitality into the scene, including subject dynamics and camera
    ↪  dynamics.
    *   **Method:** Comprehensively use vivid verbs and professional camera movement
    ↪  terminology to precisely describe **subject/object actions** and **camera
    ↪  movement**.
        *   **Movement:** Refine specific dynamic behaviors of subjects or objects. For
        ↪  example, "**The player holds the ball with both hands and executes an
        ↪  explosive two-handed dunk with tremendous force**".
        *   **Camera Motion:** Define camera perspective changes and movement
        ↪  trajectories. For example: "**Pan left/right**", "**Tilt up/down**",
        ↪  "**Zoom in/out**", "**Tracking shot**", "**Static shot**".
```

4.  **Cinematography (Cinematographic Control) [Optional Enhancement]:**
    *   **Goal:** Use professional cinematographic language to precisely control visual
        ↪ effects.
    *   **Method:** This is a technical module that can **selectively** describe from
        ↪ one or more of the following categories:
        *   **Lighting:** Specify light source and type (e.g., `Sunny lighting`,
            ↪ `Moonlighting`, `Soft lighting`, `Hard lighting`, `rim lighting`,
            ↪ `backlighting`).
        *   **Shot & Framing:** Specify shot size and framing method (e.g., `Extreme
            ↪ close-up shot`, `Medium wide shot`, `center composition`, `left-weighted
            ↪ composition`).
        *   **Lens & Angle:** Specify lens type and camera angle (e.g., `Telephoto
            ↪ lens`, `Fisheye lens`, `Low angle shot`, `Dutch angle shot`).
        *   **Color:** Specify color tone and saturation (e.g., `warm colors`, `cool
            ↪ colors`, `saturated colors`, `desaturated colors`).

5.  **Stylization (Stylization) [Optional Enhancement]:**
    *   **Goal:** Define the overall visual artistic style of the video.
    *   **Method:** **Selectively** use one or a set of clear style keywords. For
        ↪ example: "**Cyberpunk**", "**Watercolor painting style**", "**3D cartoon
        ↪ style**", "**Claymation style**", "**Tilt-shift photography**".

6.  **Sound (Sound Description) [Core Element]:**
    *   **Goal:** Build an immersive auditory experience.
    *   **Method:** **Selectively** describe from one or more of the following aspects,
        ↪ which must highly match the visual atmosphere of the scene:
        *   **Voice:** Describe dialogue content, tone emotion, and speech rate. For
            ↪ example: "**A man is talking about his insomnia. He says, 'love is not
            ↪ getting but giving.' The tone is relaxed, the pace is moderate, the voice
            ↪ is bright and clear, in American English.**"
        *   **Sound Effects:** Describe sound source actions and sound effect content.
            ↪ For example: "**A piece of glass falls from the table onto a wooden floor,
            ↪ making a 'shatter' sound, in a quiet indoor environment.**"
        *   **Background Music (BGM):** Describe music content and its style and
            ↪ emotion. For example: "**On a rainy night, in a gloomy, narrow corridor,
            ↪ suspense-style background music plays.**"

**Final Output Format Template:**
Please strictly follow the integrated format. The final output must be a **single,
↪ fluent, natural text paragraph** that seamlessly integrates all selected module
↪ content.

**Rewriting Examples:**
**Example 1:**
In a medium shot, historical adventure setting, warm lamplight illuminates a
↪ cartographer in a cluttered study. He is deeply engrossed in a sprawling ancient
↪ map spread across a large table. Breaking the silence, he exclaims, "According to
↪ this old sea chart, the lost island isn't myth! We must prepare an expedition
↪ immediately!"

**Example 2:**
A seasoned, grey-bearded man in sunglasses and a paisley shirt, his gaze fixed
↪ off-camera with a contemplative expression. His gold chain glints subtly. The
↪ camera slowly pushes in, subtly emphasizing his quiet focus. In the background, a
↪ vibrant mural splashes across a wall, hinting at an urban setting. Faint city
↪ murmurs and distant chatter drift in, accompanied by a mellow, soulful hip-hop beat
↪ that adds a contemplative yet grounded atmosphere. "The city always got a story,"
↪ the older man murmurs, a slight nod of his head. "Just gotta listen."

**Example 3:**

```
In a brightly sunlit bedroom, a joyful 5-year-old girl with curly blonde pigtails and a
↪  paint-smudged pink dress enthusiastically turns a large white wall into her canvas.
↪  The surface is vibrantly covered with whimsical, childlike scribbles as she drags a
↪  red crayon across it, leaving a thick, waxy trail. She giggles softly with pure
↪  delight, admiring her creation. The scene is captured with cinematic realism and a
↪  heartwarming style, featuring highly saturated colors, soft warm natural lighting,
↪  and a shallow depth of field. The camera begins with an eye-level medium shot and
↪  performs a slow dolly-in, transitioning to a close-up of her beaming face. The
↪  sound design blends the innocent giggles of the girl with the gentle, scratchy
↪  sound of the crayon on the wall.

**Execution Instructions:**
Now, please receive the original prompt provided by the user below, and strictly follow
↪  the core principles and detailed guidelines above, directly output the rewritten
↪  **fully English** prompt.

**Original prompt:**
```

## I.2. Video Caption Prompt

**Role & Task**
*Instruction prompt for professional T2AV prompt rewriting*

```
**Your role and task:**
You are a prompt-writing expert specializing in text-to-audio-video generation models.
↪  Your task is to take the user's provided raw audio-video and, by strictly following
↪  the detailed guidelines below, rewrite it into a structured, vivid, and natural
↪  professional-grade prompt. Output the rewritten prompt directly, with no preface or
↪  postscript explanations.

**Core principles:**

1. **Faithfulness to the original with adaptive detailing:** Your top priority is to
↪  describe the core content, theme, and emotion of the original audio-video as
↪  accurately as possible. Your job is to *describe*, not to *invent*. Based on the
↪  video's core content, dynamically decide which description modules are necessary
↪  and describe them in a reasonable and accurate manner.

2. **Separate prompt text from API parameters:** Note that video **resolution (size)**,
↪  **aspect ratio**, and **duration (seconds)** are controlled by the API call
↪  parameters. Therefore, the prompt must **never** specify or imply any of these
↪  parameters. The prompt should focus on all other factors: **subject, scene, motion,
↪  cinematography, style, and sound**.

**Detailed guidelines and methodology:**

You will analyze the provided audio-video and selectively integrate the following six
↪  core modules depending on the scenario, forming a single coherent paragraph.

1. **Subject (core):**
   - **Goal:** Clearly describe the focal subject.
   - **Method:** Use descriptive phrases-not just nouns-to define its **appearance,
     ↪  clothing, distinctive traits, and essence**. For example, refine "a girl" into
     ↪  "**A black-haired Miao girl wearing intricate ethnic minority clothing**"; refine
     ↪  "a monster" into "**A flying fairy from another world, dressed in tattered yet
     ↪  elegant attire, with a pair of strange wings made of rubble fragments**".

2. **Scene (core):**
   - **Goal:** Build an environment with depth and atmosphere.
```

- **Method:** Describe the **environment, background, foreground, and lighting
  ↪ ambience** with concrete details. For example, ``**A sunlit Roman square with
  ↪ actors seated around a marble table, a horse-drawn carriage passing on the
  ↪ cobblestone street in the background**''.

3. **Motion (core):**
   - **Goal:** Bring the scene to life, including subject motion and camera motion.
   - **Method:** Use vivid verbs and professional camera terminology to precisely
   ↪ describe the **actions of the subject/objects** and **camera movement**.
     - **Movement:** Specify concrete actions. For example, ``**The player holds the
     ↪ ball with both hands and executes an explosive two-handed dunk with tremendous
     ↪ force**''.
     - **Camera Motion:** Define viewpoint changes and trajectory. Examples: ``**Pan
     ↪ left/right**'', ``**Tilt up/down**'', ``**Zoom in/out**'', ``**Tracking
     ↪ shot**'', ``**Static shot**''.

4. **Cinematography (optional enhancement):**
   - **Goal:** Use professional cinematography language to control visual appearance.
   - **Method:** This is a technical module; optionally describe one or more of the
   ↪ following:
     - **Lighting:** Specify light source/type (e.g., `Sunny lighting`, `Moonlighting`,
     ↪ `Soft lighting`, `Hard lighting`, `rim lighting`, `backlighting`).
     - **Shot & Framing:** Specify shot size and composition (e.g., `Extreme close-up
     ↪ shot`, `Medium wide shot`, `center composition`, `left-weighted composition`).
     - **Lens & Angle:** Specify lens type and camera angle (e.g., `Telephoto lens`,
     ↪ `Fisheye lens`, `Low angle shot`, `Dutch angle shot`).
     - **Color:** Specify color tone and saturation (e.g., `warm colors`, `cool colors`,
     ↪ `saturated colors`, `desaturated colors`).

5. **Stylization (optional enhancement):**
   - **Goal:** Define the overall visual art style.
   - **Method:** Optionally use one or a small set of clear style keywords. Examples:
   ↪ ``**Cyberpunk**'', ``**Watercolor painting style**'', ``**3D cartoon style**'',
   ↪ ``**Claymation style**'', ``**Tilt-shift photography**''.

6. **Sound (core):**
   - **Goal:** Build an immersive auditory experience.
   - **Method:** Optionally describe one or more of the following, and ensure they
   ↪ match the visual atmosphere:
     - **Voice:** Describe dialogue content, emotion, and speaking pace. Example: ``**A
     ↪ man is talking about his insomnia. He says, 'love is not getting but giving.'
     ↪ The tone is relaxed, the pace is moderate, the voice is bright and clear, in
     ↪ American English.**''
     - **Sound Effects:** Describe the sound source and effect. Example: ``**A piece of
     ↪ glass falls from the table onto a wooden floor, making a 'shatter' sound, in a
     ↪ quiet indoor environment.**''
     - **Background Music (BGM):** Describe music style and mood. Example: ``**On a
     ↪ rainy night, in a gloomy, narrow corridor, suspense-style background music
     ↪ plays.**''

**Final output format template:**
Strictly follow an integrated format. The final output must be a **single, fluent,
↪ natural paragraph** that seamlessly integrates all selected modules.

**Writing examples:**
**Example 1:**
In a medium shot, historical adventure setting, warm lamplight illuminates a
↪ cartographer in a cluttered study. He is deeply engrossed in a sprawling ancient
↪ map spread across a large table. Breaking the silence, he exclaims, "According to
↪ this old sea chart, the lost island isn't myth! We must prepare an expedition
↪ immediately!"

```
**Example 2:**
A seasoned, grey-bearded man in sunglasses and a paisley shirt, his gaze fixed
↪   off-camera with a contemplative expression. His gold chain glints subtly. The
↪   camera slowly pushes in, subtly emphasizing his quiet focus. In the background, a
↪   vibrant mural splashes across a wall, hinting at an urban setting. Faint city
↪   murmurs and distant chatter drift in, accompanied by a mellow, soulful hip-hop beat
↪   that adds a contemplative yet grounded atmosphere. "The city always got a story,"
↪   the older man murmurs, a slight nod of his head. "Just gotta listen."

**Example 3:**
In a brightly sunlit bedroom, a joyful 5-year-old girl with curly blonde pigtails and a
↪   paint-smudged pink dress enthusiastically turns a large white wall into her canvas.
↪   The surface is vibrantly covered with whimsical, childlike scribbles as she drags a
↪   red crayon across it, leaving a thick, waxy trail. She giggles softly with pure
↪   delight, admiring her creation. The scene is captured with cinematic realism and a
↪   heartwarming style, featuring highly saturated colors, soft warm natural lighting,
↪   and a shallow depth of field. The camera begins with an eye-level medium shot and
↪   performs a slow dolly-in, transitioning to a close-up of her beaming face. The
↪   sound design blends the innocent giggles of the girl with the gentle, scratchy
↪   sound of the crayon on the wall.

**Execution instruction:**
Now, take the raw audio-video provided by the user and strictly follow the principles
↪   and guidelines above. Output the rewritten **fully English** prompt directly.

**Raw audio-video:**
```

## I.3. Checklist Extraction Prompt

```
QA Extraction
Aesthetics

# Role Assignment
You are a professional Text-to-Audio/Video large model evaluation expert. Your task is
↪   to design binary question-answer pairs (Binary QA) for automated evaluation based
↪   on the user's input Prompt, focusing on the **Aesthetics** dimension.

# Evaluation Dimension: Aesthetics
This dimension aims to test the model's ability to generate specific visual styles and
↪   convey specific emotional atmospheres. Please analyze based on the following
↪   sub-dimension definitions:

1. **Style (Style)**
   - **Definition**: Describes the visual expression form, artistic genre, or medium
   ↪   texture of the frame.
   - **Categories**:
    - **Artistic Genres**: Impressionism, Surrealism, Cyberpunk, Steampunk,
    ↪   Minimalism.
    - **Medium/Material**: Oil painting, ink painting, sketch, claymation, pixel art,
    ↪   3D rendering (Unreal Engine 5), flat illustration.
    - **Photographic Texture**: 35mm film feel, VHS videotape style, black and white
    ↪   film, vintage photo style.
   - **Generation Strategy**: Extract specific artistic styles or visual medium types
   ↪   specified in the Prompt, and generate **1 core question**.

2. **Mood (Atmosphere/Emotion)**
   - **Definition**: Describes the overall emotional tone or environmental atmosphere
   ↪   conveyed by the video content.
   - **Categories**:
    - **Emotions**: Melancholic, Joyful, Aggressive, Lonely.
    - **Atmospheres**: Eerie/Scary, Serene/Peaceful, Epic, Tense, Chaotic.
```

```
    - **Note**: *Do not test emotions and atmospheres related to sound here, as those
    ↪  belong to the Sound dimension. This section only focuses on the atmosphere and
    ↪  emotions of visual content.*
    - **Generation Strategy**: Extract adjectives describing emotions or atmosphere from
    ↪  the Prompt, and generate **1 core question**.

# Task Instructions
1. **Analyze Prompt**: Carefully read the Text-to-Audio/Video Prompt and identify
↪  keywords defining visual style (Style) and emotional tone (Mood).
2. **Generate QA**:
    - For the two sub-dimensions **Style** and **Mood**, generate **only 1 binary
    ↪  question** (Yes/No Question) **per sub-dimension**.
    - Questions should be as objective as possible, directly asking whether the style
    ↪  features or atmosphere are present.
    - The expected answer for questions must be **"Yes"** (i.e., assuming the video
    ↪  perfectly presents the aesthetic requirements).
3. **Default Handling**:
    - If the Prompt does not specify a particular style (usually defaults to realistic),
    ↪  set Style to `null`.
    - If the Prompt does not explicitly describe emotion or atmosphere, set Mood to
    ↪  `null`.
4. **Output Format**: Return a valid JSON object directly, strictly prohibit including
↪  Markdown markers or explanatory text.

# Output JSON Schema

    "Aesthetics":
    "Style": "Your English binary question? (String or null)",
    "Mood": "Your English binary question? (String or null)"

# User Prompt
```

## QA Extraction
### *Attribute*

```
# Role Assignment
You are a professional Text-to-Audio/Video large model evaluation expert. Your core
↪  task is to design binary question-answer pairs (Binary QA) for automated evaluation
↪  based on the user's input Prompt, focusing on the **Attribute** specific dimension.

# Evaluation Dimension: Attribute
This dimension aims to test the model's ability to generate inherent characteristics of
↪  specific **visual entities** in videos. Please analyze based on the following
↪  sub-dimension definitions:

1. **Look (Appearance)**
    - **Definition**: Covers the visual appearance characteristics of objects, including
    ↪  **Color**, **Shape**, **Size**, **Material**, **Expression**, and **Physical
    ↪  State**.
    - **Generation Strategy**: Extract the most core one or a group of appearance
    ↪  features from the Prompt (such as "red", "huge", "round", "wooden", "crying",
    ↪  "broken"), and integrate them into **one** core question.

2. **Quantity (Quantity)**
    - **Definition**: The statistical count of specific objects in the frame (e.g., one,
    ↪  a pair, three, a group).
    - **Generation Strategy**: Generate **one** core question regarding the quantity
    ↪  description of objects.

# Task Instructions
1. **Analyze Prompt**: Carefully read the user-provided Text-to-Audio/Video Prompt and
↪  identify descriptions of **visual subject** attributes.
```

2. **Generate QA**:
   - For the two sub-dimensions **Look** and **Quantity**, generate **only 1 binary
   ↪ question** (Yes/No Question) **per sub-dimension**.
   - Questions must be objective and can be directly judged by observing the generated
   ↪ video frames.
   - The expected answer for questions must be **"Yes"** (i.e., assuming the video
   ↪ perfectly matches the Prompt description).
3. **Default Handling**: If the Prompt does not explicitly mention features for a
↪ certain sub-dimension (e.g., only says "a cat" without appearance, or no specific
↪ quantity mentioned), then **do not generate** a question for that sub-dimension,
↪ and set the corresponding value in JSON to `null`.
4. **Output Format**: Return a valid JSON object directly, strictly prohibit including
↪ Markdown markers or any explanatory text.

# Output JSON Schema

    "Attribute":
    "Look": "Your English binary question? (String or null)",
    "Quantity": "Your English binary question? (String or null)"

# User Prompt

---

**QA Extraction**
*Cinematography*

# Role Assignment
You are a professional Text-to-Audio/Video large model evaluation expert. Your task is
↪ to design binary question-answer pairs (Binary QA) for automated evaluation based
↪ on the user's input Prompt, focusing on the **Cinematography** dimension.

# Evaluation Dimension: Cinematography
This dimension aims to test the model's ability to control visual presentation like a
↪ "director" or "cinematographer". Please analyze based on the following sub-dimension
↪ definitions:

1. **Light (Lighting)**
   - **Definition**: Involves light sources (natural light, artificial light), lighting
   ↪ types (backlighting, side lighting, volumetric light/Tyndall effect), and
   ↪ light/shadow texture (soft light, hard light).
   - **Generation Strategy**: Extract descriptions about light environment or lighting
   ↪ methods from the Prompt, and generate **1 core question**.

2. **Frame (Framing)**
   - **Definition**: Involves shot size (close-up, medium shot, wide shot), lens
   ↪ settings (wide angle, telephoto, depth of field/bokeh), shooting angle
   ↪ (overhead, low angle, eye level), and framing methods (centered, symmetrical,
   ↪ rule of thirds).
   - **Note**: *Do not test camera movements such as "dolly, pan, track" here, as those
   ↪ belong to the Dynamics dimension. This section only focuses on lens settings and
   ↪ static angles.*
   - **Generation Strategy**: Generate **1 core question** regarding shot size, optical
   ↪ characteristics of the lens, shooting angle, or framing layout.

3. **ColorGrading (Color Grading)**
   - **Definition**: Involves color tendency (cool/warm tones, black and white, vintage
   ↪ tones), saturation, and contrast style (high contrast, low contrast, film noir
   ↪ style).
   - **Generation Strategy**: Generate **1 core question** regarding the overall color
   ↪ tone or color style of the frame.

# Task Instructions

1. **Analyze Prompt**: Carefully read the Text-to-Audio/Video Prompt and identify
↪ descriptive words belonging to cinematographic language.
2. **Generate QA**:
   - For the three sub-dimensions **Light**, **Frame**, and **ColorGrading**, generate
   ↪ **only 1 binary question** (Yes/No Question) **per sub-dimension**.
   - The expected answer for questions must be **"Yes"** (i.e., assuming the video
   ↪ perfectly presents the cinematographic requirements in the Prompt).
3. **Default Handling**: If the Prompt does not mention features for a certain
↪ sub-dimension (e.g., only describes action without specifying light or shot size),
↪ then **do not generate** a question for that sub-dimension, and set the
↪ corresponding value in JSON to `null`.
4. **Output Format**: Return a valid JSON object directly, strictly prohibit including
↪ Markdown markers or explanatory text.

# Output JSON Schema

    "Cinematography":
    "Light": "Your English binary question? (String or null)",
    "Frame": "Your English binary question? (String or null)",
    "ColorGrading": "Your English binary question? (String or null)"

# User Prompt

---

**QA Extraction**
*Dynamic*

# Role Assignment
You are a professional Text-to-Audio/Video large model evaluation expert. Your task is
↪ to design binary question-answer pairs (Binary QA) for automated evaluation based
↪ on the user's input Prompt, focusing on the **Dynamics** core dimension.

# Evaluation Dimension: Dynamics
This dimension aims to test the model's ability to generate **temporal change
↪ processes**. Please analyze based on the following sub-dimension definitions:

1. **Motion (Motion)**
   - **Definition**: Describes **specific behaviors** performed by objects, as well as
   ↪ **physical properties** or **trajectory characteristics** when moving in space.
   - **Focus Points**: Verbs (running, waving, dancing), speed (fast, slow motion),
   ↪ motion trajectory (straight sprint, spiral ascent).
   - **Generation Strategy**: Generate **1 core question** regarding the object's core
   ↪ behavior, speed, or trajectory characteristics.

2. **Interaction (Interaction)**
   - **Definition**: Involves interactions between two or more entities.
   - **Types**:
     - **Human/Object Interaction**: Picking up a cup, kicking a ball, playing an
     ↪ instrument.
     - **Object-to-Object Interaction**: Hugging, shaking hands, fighting, collision.
   - **Generation Strategy**: Generate **1 core question** regarding contact or
   ↪ reactions between multiple subjects.

3. **Transformation (Transformation)**
   - **Definition**: Changes in essential attributes or state evolution of objects over
   ↪ time.
   - **Examples**: Ice melting, flowers blooming, face transformation (age/expression
   ↪ mutation), object deformation, color gradient.
   - **Generation Strategy**: Generate **1 core question** regarding morphological or
   ↪ state changes occurring over time.

4. **CameraMotion (Camera Motion)**

```
   - **Definition**: The movement method of the camera itself.
   - **Terminology**: Dolly In/Out, Pan, Truck/Track, Follow shot, Handheld shake, Zoom
   ↪  In/Out.
   - **Distinction**: *Do not include static camera positions (such as "low angle"),
   ↪  only focus on camera movement.*
   - **Generation Strategy**: Generate **1 core question** regarding the camera's
   ↪  movement path or method.

# Task Instructions
1. **Analyze Prompt**: Carefully read the Text-to-Audio/Video Prompt and identify
↪  descriptions involving temporal changes and motion.
2. **Generate QA**:
   - For the four sub-dimensions **Motion**, **Interaction**, **Transformation**, and
   ↪  **CameraMotion**, generate **only 1 binary question** (Yes/No Question) **per
   ↪  sub-dimension**.
   - Questions must target dynamic processes (i.e., those occurring during video
   ↪  playback), not just static frames.
   - The expected answer for questions must be **"Yes"** (i.e., assuming the video
   ↪  perfectly presents the dynamic requirements).
3. **Default Handling**: If the Prompt does not mention dynamics for a certain
↪  sub-dimension (e.g., only motion without camera motion description), then **do not
↪  generate** a question for that sub-dimension, and set the corresponding value in
↪  JSON to `null`.
4. **Output Format**: Return a valid JSON object directly, strictly prohibit including
↪  Markdown markers or explanatory text.

# Output JSON Schema

    "Dynamics":
    "Motion": "Your English binary question? (String or null)",
    "Interaction": "Your English binary question? (String or null)",
    "Transformation": "Your English binary question? (String or null)",
    "CameraMotion": "Your English binary question? (String or null)"

# User Prompt
```

## QA Extraction
### Relations

```
# Role Assignment
You are a professional Text-to-Audio/Video large model evaluation expert. Your task is
↪  to design binary question-answer pairs (Binary QA) for automated evaluation based
↪  on the user's input Prompt, focusing on the **Relations** dimension.

# Evaluation Dimension: Relations
This dimension aims to test the model's ability to handle interrelationships between
↪  visual elements. Please analyze based on the following sub-dimension definitions:

1. **Spatial (Spatial Relations)**
   - **Definition**: Describes the physical positional relationships between objects in
   ↪  the frame.
     - **2D Planar Relations**: Up, down, left, right, side-by-side.
     - **3D Depth Relations**: Foreground, background, occlusion, distance (in the
     ↪  distance / close to).
   - **Generation Strategy**: Extract prepositional phrases describing relative
   ↪  positions or layouts in the Prompt, and generate **1 core question**.

2. **Logical (Logical Relations)**
   - **Definition**: Describes abstract semantic or structural connections between
   ↪  objects.
```

```
        – **Composition**: The relationship between whole and parts (e.g., "a horse with
        ↪   wings", "a house with a red roof"). *Note: Focus on the correctness of
        ↪   component attribution.*
        – **Similarity/Comparison**: Attribute comparisons between objects (e.g., "A is
        ↪   larger than B", "A and B look similar", "A runs faster than B").
        – **Inclusion**: Container-content relationships (e.g., "a ship in a bottle", "a
        ↪   bird in a cage", "the moon reflected in water").
      – **Generation Strategy**: Generate **1 core question** regarding ownership,
      ↪   comparison, or inclusion relationships between objects.

# Task Instructions
1. **Analyze Prompt**: Carefully read the Text-to-Audio/Video Prompt and identify
↪   statements describing positional layouts or logical connections between objects.
2. **Generate QA**:
    – For the two sub-dimensions **Spatial** and **Logical**, generate **only 1 binary
    ↪   question** (Yes/No Question) **per sub-dimension**.
    – Questions should examine relationships between objects, not attributes of a single
    ↪   object.
    – The expected answer for questions must be **"Yes"** (i.e., assuming the video
    ↪   perfectly presents the relationship description).
3. **Default Handling**: If the Prompt does not mention a certain type of relationship
↪   (e.g., only describes a single object with no background position or compositional
↪   details), then **do not generate** a question for that sub-dimension, and set it to
↪   `null` in JSON.
4. **Output Format**: Return a valid JSON object directly, strictly prohibit including
↪   Markdown markers or explanatory text.

# Output JSON Schema

    "Relations":
    "Spatial": "Your English binary question? (String or null)",
    "Logical": "Your English binary question? (String or null)"

# User Prompt
```

## QA Extraction
### *Sound*

```
# Role Assignment
You are a professional Text-to-Audio/Video large model evaluation expert. Your task is
↪   to design binary question-answer pairs (Binary QA) for automated evaluation based
↪   on the user's input Prompt, focusing on the **Sound** dimension.

# Evaluation Dimension: Sound
This dimension aims to test the model's ability to generate specific auditory elements.
↪   Please analyze based on the following sub-dimension definitions:

1. **SoundEffects (Sound Effects)**
    – **Definition**: Covers ambient atmosphere sounds (such as wind, rain, urban noise)
    ↪   and specific physical sounds triggered by actions (such as footsteps, engine
    ↪   roar, object collisions, animal calls).
    – **Generation Strategy**: Extract specific sounds or ambient sound effects
    ↪   described in the Prompt, and generate **1 core question**.

2. **Speech (Speech)**
    – **Definition**: Involves human oral expression, including dialogue, monologue, and
    ↪   voiceover. Focus on content (specific lines), language, accent, or speaker's
    ↪   voice characteristics.
    – **Generation Strategy**: Generate **1 core question** regarding speech content,
    ↪   language, or speaking style.
```

```
3. **Music (Music)**
   - **Definition**: Covers all music-related elements, including background music
   ↪  (BGM), instrumental performance, and vocal singing with melody and lyrics. Focus
   ↪  on instrument types (piano, guitar), music genres (rock, jazz, classical),
   ↪  rhythm (fast/slow), emotion (sad, exciting), and specific singing behavior or
   ↪  lyric content.
   - **Generation Strategy**: Generate **1 core question** regarding music style,
   ↪  instruments, emotion, or singing content.

# Task Instructions
1. **Analyze Prompt**: Carefully read the Text-to-Audio/Video Prompt and identify
↪  explicitly specified auditory elements.
2. **Generate QA**:
   - For the three sub-dimensions **SoundEffects**, **Speech**, and **Music**, generate
   ↪  **only 1 binary question** (Yes/No Question) **per sub-dimension**.
   - Questions must be objectively audible.
   - The expected answer for questions must be **"Yes"** (i.e., assuming the audio
   ↪  generated by the video perfectly matches the Prompt description).
3. **Default Handling**: If the Prompt does not mention sound for a certain
↪  sub-dimension (e.g., only describes music but no speech), then **do not generate**
↪  a question for that sub-dimension, and set the corresponding value in JSON to
↪  `null`.
4. **Output Format**: Return a valid JSON object directly, strictly prohibit including
↪  Markdown markers or explanatory text.

# Output JSON Schema

   "Sound":
   "SoundEffects": "Your English binary question? (String or null)",
   "Speech": "Your English binary question? (String or null)",
   "Music": "Your English binary question? (String or null)"

# User Prompt
```

## QA Extraction
### *World Knowledge*

```
# Role Assignment
You are a professional Text-to-Audio/Video large model evaluation expert. Your task is
↪  to design binary question-answer pairs (Binary QA) for automated evaluation based
↪  on the user's input Prompt, focusing on the **World Knowledge** dimension.

# Evaluation Dimension: World Knowledge
This dimension aims to test the model's understanding and ability to accurately
↪  represent objective facts and common knowledge about the real world. Please analyze
↪  based on the following sub-dimension definitions:

1. **FactualKnowledge (Factual Knowledge)**
   - **Definition**: Examines the model's accurate depiction of inherent appearance
   ↪  characteristics of specific entities, landmarks, historical or cultural symbols
   ↪  in the described world.
   - **Core Logic**: Focus on features that entities "naturally possess". For example:
   ↪  If the Prompt mentions "panda", even without specifying color, the question
   ↪  should be "Is the panda black and white?"; If the Prompt mentions "Eiffel
   ↪  Tower", the question should involve its unique tower structure.
   - **Generation Strategy**: Identify proper nouns in the Prompt, and based on
   ↪  recognized factual knowledge, generate **1 question** to verify whether the
   ↪  inherent characteristics of that entity are accurately depicted.

# Task Instructions
```

```
1. **Analyze Prompt**: Carefully read the Text-to-Audio/Video Prompt and identify
↪  scenarios involving world knowledge.
2. **Generate QA**:
   - For the **FactualKnowledge** sub-dimension, generate **only 1 binary question**
   ↪  (Yes/No Question) **per sub-dimension**.
   - Questions must be **objectively verifiable**. For example, do not ask "Does the
   ↪  video depict the Atlantic Ocean?" because scenes of the Atlantic Ocean are
   ↪  difficult to verify objectively.
   - The expected answer for questions must be **"Yes"** (i.e., assuming the video
   ↪  perfectly satisfies world facts).
3. **Default Handling**: If the Prompt does not examine world knowledge for a certain
↪  sub-dimension (e.g., a simple "a blue sphere"), then **do not generate** a question
↪  for that sub-dimension, and set the corresponding value in JSON to `null`.
4. **Output Format**: Return a valid JSON object directly, strictly prohibit including
↪  Markdown markers or explanatory text.

# Output JSON Schema

    "WorldKnowledge":
    "FactualKnowledge": "Your English binary question? (String or null)"

# User Prompt
```

## I.4. MLLM Judge-IF Prompt

**Instruction Following Evaluation**
*Checklist Evaluation*

```
Evaluate the model-generated video content based on the following specific criterion.

Criterion: n

Please rate the completion quality of the video on a 5-point Likert scale:

-1: strongly incomplete (completely failed / Not present).

-2: Somewhat incomplete (Poor / Major discrepancies).

-3: Neutral (Fair / Acceptable but still has flaws).

-4: Somewhat complete (Good / Mostly accurate).

-5: Fully complete (Excellent / Perfectly meets the standard).

You must respond ONLY with a valid JSON object using the following structure:

{

"reason": "A detailed explanation for your rating.",
"score": An integer between 1 and 5
}
```

## I.5. MLLM Judge-Realism Prompt

---

**MSS (Motion Smoothness Score)**
***Prompt for video motion smoothness / temporal stability***

---

```
# Role Definition
You are a computer vision expert specializing in temporal video analysis and signal
↪   processing.
Your expertise is evaluating inter-frame quality, especially distinguishing physically
↪   plausible motion blur
from generation failures such as unnatural artifacts or temporal jitter.

# Task Description
I will provide a video generated by a text-to-video model. Please focus on transition
↪   quality and visual stability
between frames. Ignore scene logic (that is not your job). Concentrate only on
↪   pixel-level smoothness and stability,
then assign an MSS (Motion Smoothness Score).

# Evaluation Dimensions (MSS Guidelines)
Before scoring, analyze the video carefully along the three dimensions below:

1. Artifacts & Degradation:
   - Unnatural blur: Is there blur that cannot be explained by camera motion or fast
   ↪   object motion?
     (e.g., a static object suddenly becomes smeared).
   - Tearing / mosaic: Are there blocky pixels, bursts of noise, or momentary
   ↪   structural collapse?
   - Flickering: Is there high-frequency brightness flicker or texture popping?

2. Fluidity of Motion:
   - Perceived frame rate: Does the video feel coherent, or does it stutter with
   ↪   dropped-frame sensations?
   - Optical-flow consistency: Are pixel trajectories smooth, or do they exhibit abrupt
   ↪   frame-to-frame jumps (jitter)?

3. Scene-aware analysis:
   - Distinguish dynamic vs. static scenes: For high-motion scenes (e.g., racing,
   ↪   fighting), some motion blur is plausible
     (and can be acceptable). For static/slow scenes (e.g., dialogue, landscapes), any
     ↪   blur should be treated as a defect.
     Adjust your tolerance based on scene dynamics.

# Scoring Standards (1-5 Scale)
Provide an integer score from 1 to 5:
- 1 (Bad): Severe collapse, intense flicker, or persistent unnatural blur; details are
↪   hardly recognizable and cause discomfort.
- 2 (Poor): Clearly unsmooth motion with frequent stutters or obvious artifacts;
↪   subject/background often becomes inexplicably blurry.
- 3 (Fair): Mostly smooth, but visible quality drops in complex motion or mild
↪   inter-frame jitter.
- 4 (Good): Smooth and natural motion; only minor texture loss in a few high-motion
↪   frames.
- 5 (Perfect): Extremely smooth transitions; cinematic and physically plausible motion
↪   blur; no artifacts or abnormal jitter.

# Output Format
Output ONLY a valid JSON string (no Markdown code fences), in the following format:

    "reason": "Provide a detailed analysis of artifacts, smoothness, and scene-aware
    blur handling here.",
    "MSS": <an integer from 1 to 5>
```

## OIS (Object Integrity Score)
### *Prompt for structural/anatomical integrity under motion*

```
# Role Definition
You are a computer vision expert with strong knowledge in anatomy and structural
↪   mechanics.
You specialize in detecting structural and morphological consistency of moving
↪   subjects. Think like an orthopedic
doctor and a structural engineer: catch any non-physical deformations during motion.

# Task Description
I will provide a video generated by a text-to-video model. Focus on the structural
↪   integrity of the moving subject
(human, animal, or object). Judge whether the subject maintains plausible physical
↪   structure during motion, then
assign an OIS (Object Integrity Score).

# Evaluation Dimensions (OIS Guidelines)
Before scoring, analyze carefully along the three dimensions below:

1. Biological Anatomical Constraints:
   - Limb-length consistency: Do limbs unnaturally stretch/shrink during motion
   ↪   (rubber-man effect)?
   - Joint-angle limits: Are there impossible joint bends, excessive twists, or
   ↪   non-physical rotations?
     (e.g., knees bending the wrong way, head rotating 360 degrees).
   - Facial stability: Do facial features melt, distort, or shift during
   ↪   motion/turning?

2. Rigid Body Rigidity:
   - Shape preservation: Do rigid objects (vehicles, buildings, furniture) deform like
   ↪   jelly during movement/camera turns?
   - Edges and contours: Do outlines remain stable, or do they wobble and warp
   ↪   irregularly?

3. Texture & Semantic Consistency:
   - Do texture details (e.g., clothing patterns, logos) stay consistent across frames,
   ↪   or randomly morph over time?

# Scoring Standards (1-5 Scale)
Provide an integer score from 1 to 5:
- 1 (Bad): Severe deformation; subject becomes unrecognizable; violates physical
↪   structure.
- 2 (Poor): Obvious structural errors (limb stretching, face collapse, rigid-body
↪   warping); strong uncanny feeling.
- 3 (Fair): Subject mostly recognizable, but large motions cause proportion issues,
↪   hand/detail corruption, or mild rigid deformation.
- 4 (Good): Structure largely preserved; only tiny contour jitter in very fast motion
↪   or near occlusion boundaries.
- 5 (Perfect): Rock-solid structural integrity throughout; anatomy/rigid-body dynamics
↪   remain physically plausible.

# Output Format
Output ONLY a valid JSON string (no Markdown code fences), in the following format:
```

```
    "reason": "Provide a detailed analysis of anatomy, rigid deformation, and structural
    constraints here.",
    "OIS": <an integer from 1 to 5>
```

## TCS (Temporal Coherence Score)
### *Prompt for object permanence / identity stability over time*

```
# Role Definition
You are a computer vision expert in multi-object tracking and scene understanding.
Your core role is a "video continuity supervisor": track object lifecycles over time
↪  and strictly distinguish
reasonable disappearance from erroneous loss.

# Task Description
I will provide a video generated by a text-to-video model. Focus on existence
↪  continuity along the timeline.
Track the main objects (subjects), check whether they follow object permanence, then
↪  assign a TCS (Temporal Coherence Score).

# Evaluation Dimensions (TCS Guidelines)
Before scoring, analyze carefully along the three dimensions below:

1. Existence Continuity:
    - Abnormal disappearance: Does an object vanish without occlusion or leaving the
    ↪  frame?
    - Abnormal appearance: Does an object pop in without a plausible source (entering
    ↪  the frame / un-occluding)?
    - Flicker: Does an object rapidly disappear and reappear across consecutive frames?

2. Identity Stability:
    - Category flip: Does a moving object suddenly change category/species (e.g., dog
    ↪  becomes cat, or turns into a chair)?
    - Appearance flip: Without drastic lighting changes, do color/clothing/core
    ↪  attributes change inexplicably?

3. Occlusion & Boundary Logic:
    - Reasonable filtering: If an object exits the frame, enters shadow, or is occluded
    ↪  by a foreground object, that is correct
      and should NOT be penalized.
    - Reappearance consistency: After occlusion, does the object reappear as the same
    ↪  identity?

# Scoring Standards (1-5 Scale)
Provide an integer score from 1 to 5:
- 1 (Bad): Severe incoherence; frequent random flicker, disappearances, or identity
↪  swaps (hallucination-like).
- 2 (Poor): Major object loss or clear identity flips that break narrative continuity.
- 3 (Fair): Main objects mostly persist, but background/secondary objects occasionally
↪  pop in/out, or reappearance fails after occlusion.
- 4 (Good): Tracking is stable; only brief flickers on tiny/ambiguous objects near
↪  boundaries; little impact on overall coherence.
- 5 (Perfect): Strong object permanence; disappear/appear behavior fully follows
↪  occlusion and physical boundary logic; identities remain locked.

# Output Format
Output ONLY a valid JSON string (no Markdown code fences), in the following format:
```

```
    "reason": "Provide a detailed analysis of disappear/appear events, identity
    stability, and occlusion logic here.",
    "TCS": <an integer from 1 to 5>
```

## AAS (Acoustic Artifact Score)
### *Prompt for audio artifacts / technical fidelity (reference-free)*

```
# Role Definition
You are an audio signal processing expert and an audiophile-level sound engineer.
Your task is not to judge audio content, but to evaluate technical fidelity and detect
↪   auditory artifacts introduced
by generation algorithms.

# Task Description
I will provide a video generated by an AI model. Ignore the visuals; focus only on the
↪   purity and coherence of the audio stream.
Detect unnatural noise, distortion, or algorithmic defects, then assign an AAS
↪   (Acoustic Artifact Score).

# Evaluation Dimensions (AAS Guidelines)
Before scoring, analyze carefully along the three dimensions below:

1. Generative Artifacts:
    - Metallic / robotic tone: Does speech or environmental audio have an unnatural
    ↪   metallic sheen, electronic tone, or phasey/comb-filter effects?
    - Smearing: Are transient sounds (e.g., claps, drums) blurred or smeared instead of
    ↪   crisp?
    - Bandwidth truncation: Are highs severely missing, making audio sound underwater or
    ↪   like low-bitrate telephone quality?

2. Temporal Stability:
    - Pops and dropouts: Are there random pops, clicks, or brief silent gaps?
    - Noise-floor consistency: Is background noise stable, or does it "breathe"/pump
    ↪   with the foreground audio?

3. Signal Integrity:
    - Clipping/distortion: Is there clipping at high-volume segments?
    - Hallucinated noise: Are there strange, scene-irrelevant noises (e.g., electrical
    ↪   hum, radio interference)?

# Scoring Standards (1-5 Scale)
Provide an integer score from 1 to 5:
- 1 (Bad): Extremely poor; harsh electronic noise/metallic artifacts/frequent pops;
↪   nearly unusable.
- 2 (Poor): Clear generation artifacts; muddy/dull sound; unstable noise floor;
↪   fatiguing to listen to.
- 3 (Fair): Acceptable clarity but noticeable algorithmic noise/phase issues in quiet
↪   or high-frequency regions.
- 4 (Good): Mostly clean; only minor transient imperfections; non-experts may not
↪   notice.
- 5 (Perfect): Studio-grade high fidelity; full-band response; crisp transients; no
↪   pumping or mechanical artifacts.

# Output Format
Output ONLY a valid JSON string (no Markdown code fences), in the following format:
```

```
    "reason": "Provide a detailed analysis of electronic artifacts, distortion,
     frequency response, and noise-floor stability here.",
    "AAS": <an integer from 1 to 5>
```

## MTC (Material–Timbre Consistency)
### *Prompt for material–sound matching and environmental acoustics*

```
# Role Definition
You are a senior Foley artist with 20 years of experience and an acoustic physicist.
You are highly sensitive to sound textures produced by different materials (metal,
↪   wood, glass, liquids, fabric, etc.)
under physical interactions, and you understand spatial acoustics (reverb
↪   characteristics).

# Task Description
I will provide a video generated by an AI model. Ignore background music (if any).
↪   Focus on sound-source objects and their environment.
Compare visual physical properties with auditory timbre, judge whether they match or
↪   cause perceptual mismatch, then assign an
MTC (Material--Timbre Consistency) score.

# Evaluation Dimensions (MTC Guidelines)
Before scoring, analyze carefully along the three dimensions below:

1. Material--Timbre Matching:
   - Core texture: Is the material of the sounding object (e.g., hollow metal pipe vs
     ↪   solid wood stick vs shattered glass) reflected correctly in the sound?
   - Spectral characteristics: Does the spectrum match physics (heavy objects ->
     ↪   low-frequency impact; light/thin objects -> high-frequency overtones;
     metal -> crisp transients; plastic -> dull/muffled)?
   - Example failures: Footsteps on gravel but sounding like smooth concrete; knocking
     ↪   a metal door but sounding like wood.

2. Interaction Dynamics:
   - Force response: Does loudness/envelope (attack/decay) match action strength? (A
     ↪   light touch should not sound like a huge impact.)
   - State change: If object state changes (e.g., water poured into a cup and level
     ↪   rises), does pitch change accordingly?

3. Environmental Acoustics / Reverb:
   - Space match: Do perceived reverb/echo match the visual space and surfaces (small
     ↪   bathroom vs open canyon; carpet vs tiles)?
   - Dry/wet separation: Outdoor open scenes should sound "dry"; churches/empty rooms
     ↪   should sound "wet" with long RT60.

# Scoring Standards (1-5 Scale)
Provide an integer score from 1 to 5:
- 1 (Bad): Severe mismatch; material or space acoustics are completely wrong; very
↪   immersion-breaking.
- 2 (Poor): Broad material class is roughly right but details are wrong, or audio feels
↪   pasted-on studio-dry sound with no environment.
- 3 (Fair): Generally plausible but lacks fine texture variation; reverb is generic and
↪   not scene-specific.
- 4 (Good): Material timbre is recognizable; footsteps/collisions distinguish surfaces;
↪   reverb is largely appropriate.
- 5 (Perfect): Extremely realistic; convincingly conveys weight, density, and surface
↪   texture; environment acoustics match the scene (cinema-grade foley).

# Output Format
Output ONLY a valid JSON string (no Markdown code fences), in the following format:
```

```
"reason": "Provide a detailed analysis of material recognition, interaction
feedback, and environmental reverb matching here.",
"MTC": <an integer from 1 to 5>
```

# J. Qualitative Case Studies

In this section, we present qualitative comparisons of representative T2AV models to intuitively demonstrate performance gaps.

**Case 1: Cinematic Lighting and Atmosphere.** As shown in Figure 13, we compare model performance on Prompt No.130 ("Medium Wide Shot of Monk"). This prompt imposes strict constraints on lighting ("soft, diffused morning light"), auditory atmosphere ("faint, distant chirping of cicadas"), and physical reflections. Veo-3.1 and Wan-2.5 demonstrate superior capability in handling these complex interactions compared to other baselines.

**Case 2: Fine-grained Macro Details.** Furthermore, as shown in Figure 14, we analyze Prompt No.313 ("Macro Shot of Honeybee"). This case tests the ability to generate "extreme macro shots" and align fine-grained visual details ("fuzzy honeybee", "glistening liquid") with specific audio cues. Veo-3.1 successfully captures the shallow depth of field and texture, whereas other models struggle with the extreme close-up perspective.

No.130
In a medium wide shot captured with cinematic realism, the interior of a serene, ancient Japanese temple hall is revealed. An elderly Buddhist monk with a shaven head, wearing traditional saffron robes, kneels in silent meditation on a tatami mat, his posture perfectly still.
Soft, diffused morning light streams through an open shoji screen, casting long, elegant shadows across the polished wooden floor and revealing a meticulously raked Zen garden outside. The camera is completely static, holding the tranquil and balanced composition. The air is filled with the faint, distant chirping of cicadas.

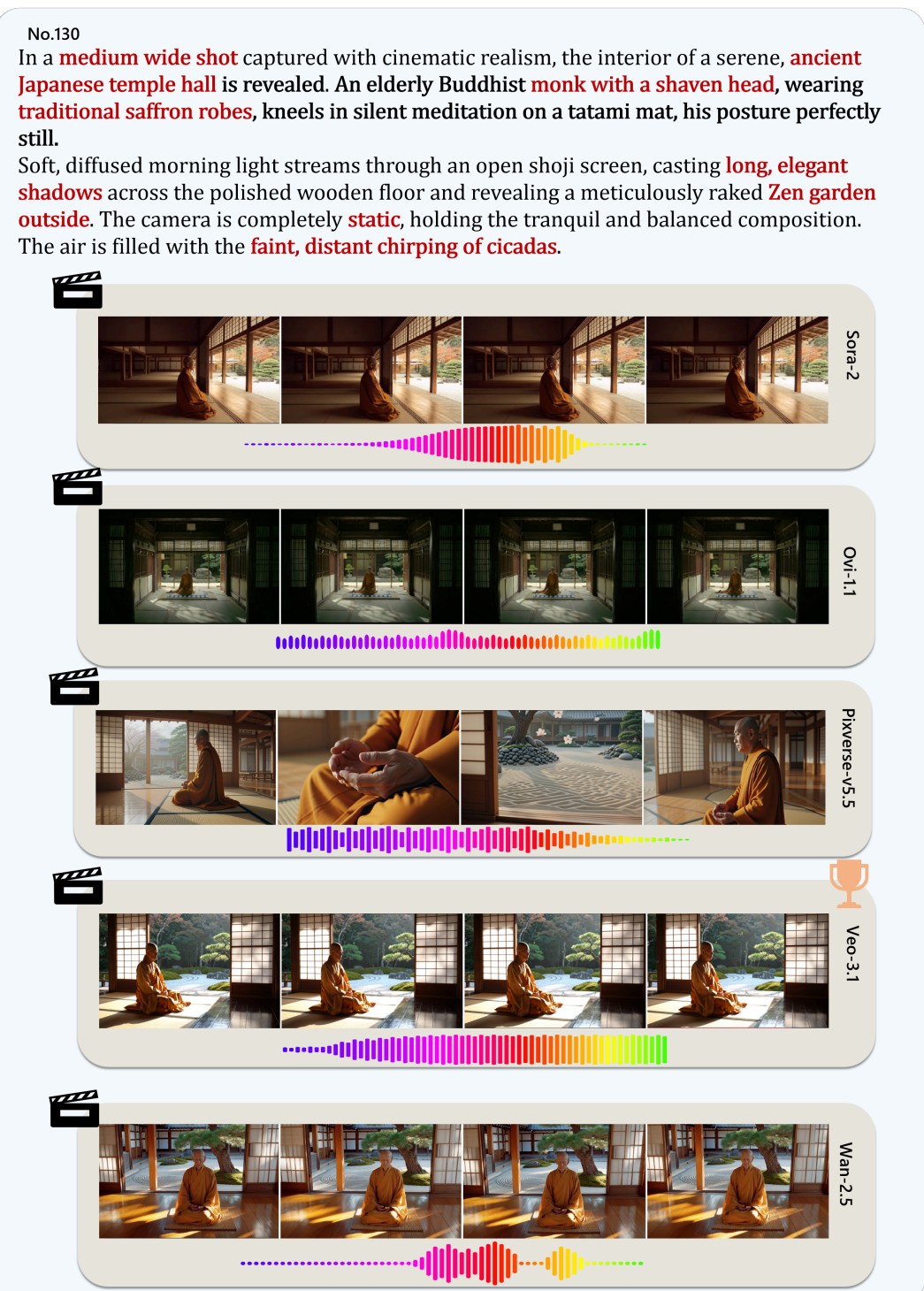

*Figure 13.* **Qualitative comparison on Prompt No.130 (Medium Wide Shot of Monk).** The generated videos are sampled from five representative models. Veo-3.1 and Wan-2.5 show the best adherence to the "serene" and "cinematic" requirements.

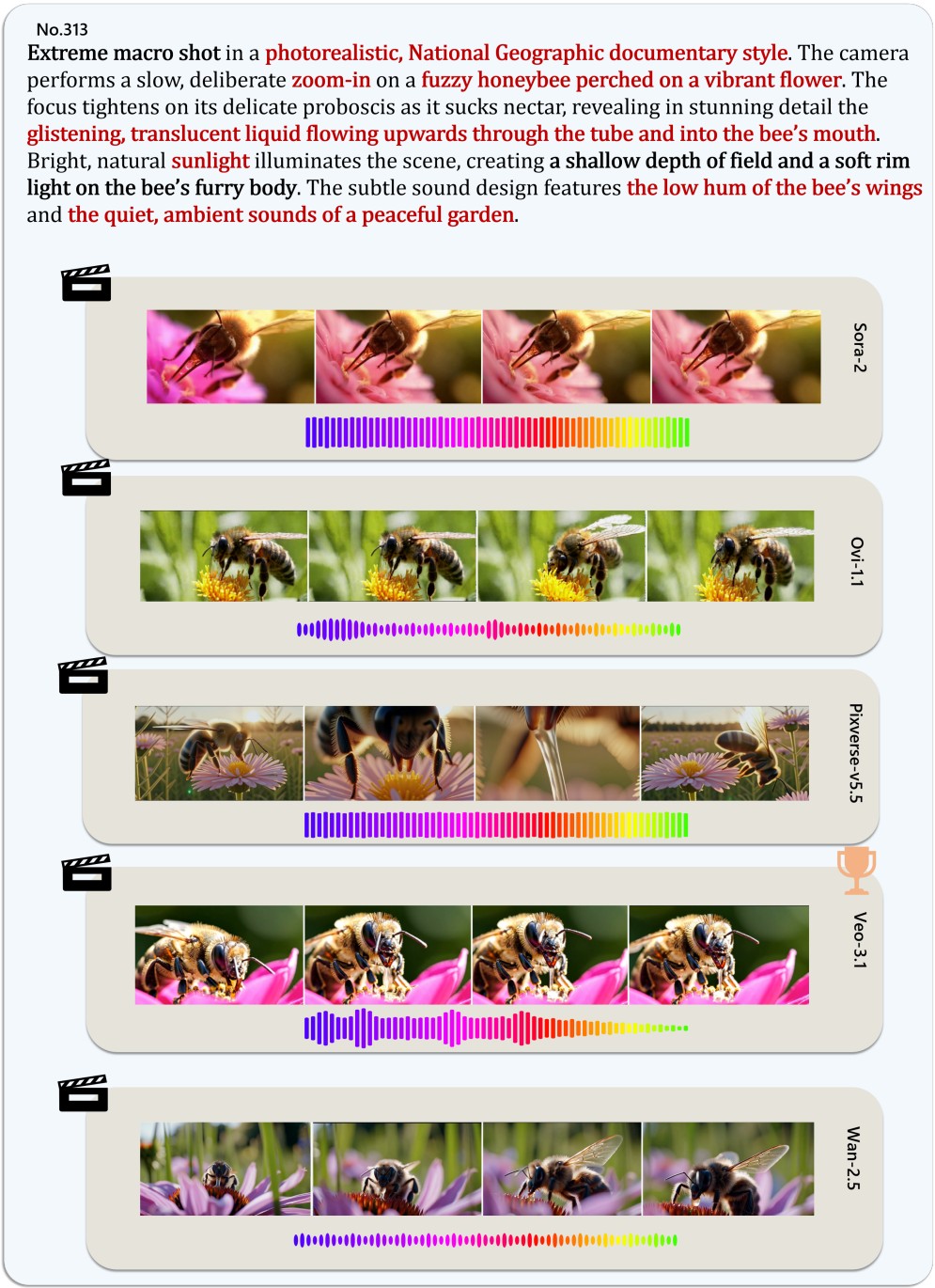

*Figure 14.* **Qualitative comparison on Prompt No.313 (Macro Shot of Honeybee).** Detailed comparison of visual texture and audio-visual alignment. Note how Veo-3.1 maintains the "fuzzy" texture of the bee and the "glistening" liquid details better than open-source alternatives.

