# OpenReview forum: "T2AV-Compass: Towards Unified Evaluation for Text-to-Audio-Video Generation"
_ICML.cc/2026/Conference — ICML 2026 regular_

### Official Review · Reviewer_YZPf · 2026-02-25

**Soundness:** 3
**Presentation:** 3
**Significance:** 3
**Originality:** 3
**Overall Recommendation:** 4
**Confidence:** 4

**Summary:**

This paper introduces T2AV-Compass, a comprehensive benchmark and evaluation framework for text-to-audio-video (T2AV) generation. It curates a set of challenging prompts and a taxonomy-driven evaluation protocol, and evaluates a wide range of representative T2AV systems using both objective metrics (covering audio quality, video quality, and cross-modal alignment) and subjective MLLM-based judging with checklist-style diagnostics. The study aims to provide a more systematic and interpretable assessment of current T2AV models, and to identify major failure modes and bottlenecks in realistic audio-visual generation.

**Compliance With Llm Reviewing Policy:**

Affirmed.

**Final Justification:**

The author has addressed most of my concerns; therefore, I would like to uphold my positive recommendation.

**Key Questions For Authors:**

Will the benchmark be released publicly? If yes, what exactly will be open-sourced, and under what license/usage restrictions?

Other concerns are detailed in the weaknesses section.

**Limitations:**

yes

**Strengths And Weaknesses:**

## Strengths

1. The benchmark targets high-complexity, real-world T2AV prompts, going beyond overly simplified settings and making the evaluation more discriminative.
2. The evaluation design is multi-dimensional, enabling a more complete characterization of system performance.
3. The subjective protocol provides diagnostic signals rather than only a single aggregate score, which is useful for model analysis and future improvements.
4. The paper evaluates a broad set of systems, including strong closed-source baselines, offering a relatively comprehensive snapshot of the current landscape.

## Weaknesses

1. The subjective judge relies on a closed-source Gemini model, which may introduce vendor/version dependency and raises questions about robustness and stability across model updates or alternative judges.
2. Although the paper states that each system is evaluated under its native configuration, differences in video resolution/frame rate and audio sample rate/loudness normalization across systems could materially affect metric outcomes and ranking fairness.
3. The paper mentions human checking verification, but it is unclear whether annotator bias is controlled or quantified. Without such analysis, it is hard to assess the reliability of human judgments and whether systematic bias may influence the reported conclusions.

---

> ### Author Rebuttal · Authors · 2026-03-31
>
> ## Q1:Subjective judge relies on a closed-source Gemini-2.5-pro.
>
> R1: We agree that dependence on a closed-source judge is an important limitation, and we do not treat Gemini-2.5-Pro as a bias-free oracle. We selected it as the primary judge because, among the candidate MLLMs tested under identical prompts and settings, it achieved the best overall agreement with human ratings across the four subjective dimensions, although Audio Realism is also its weakest dimension.
> Importantly, we did not rely only on closed-source candidates. We also experimented with an open-source judge under the same evaluation protocol, but found that its agreement with human ratings was substantially weaker than Gemini-2.5-Pro, making it less suitable as the primary judge for this benchmark. We will clarify this comparison more explicitly in the revision.
> Appendix D.3 already evaluates three candidate MLLMs under identical prompting settings, where Gemini-2.5-Pro achieves the best overall agreement with human ratings (L1 = 1.087). It also achieves system-level SRCC = 0.816 and PLCC = 0.864, compared to 0.853 / 0.886 for human-human agreement. In addition, Gemini-2.5-Pro is near human-level on Video Realism (0.937 judge-human L1 vs. 0.926 inter-human L1).
> As a robustness check, we additionally evaluated Gemini-2.5-Flash. The rankings induced by Gemini-2.5-Flash and Gemini-2.5-Pro are highly consistent across the five evaluated systems (Spearman ρ = 0.9248), with unchanged relative ordering. We will revise the paper to make this limitation and validation logic more explicit, and to clarify that the judge is used as an automatic proxy rather than a definitive oracle.
>
> ## Q2: Different model configuration leads to unstable evaluation results.
>
> R2: Our current protocol preserves each system’s native generation settings so that the benchmark reflects realistic user-facing outputs rather than an artificially normalized setup. Concretely, we keep the original frame rates and audio sampling rates.
> We already includes a repeated-run stability analysis, which shows that the judge-based evaluation is reasonably stable under fixed inputs and settings. To further probe configuration sensitivity, we conducted a post-hoc analysis on Kling-2.6’s 500 generated videos by re-encoding them under fixed codec settings while varying only video resolution (1080p / 720p / 480p) and audio sample rate (44.1 kHz / 16 kHz). The reported ranges are mean scores across these controlled conditions. We find that IF-Video and IF-Audio remain relatively stable (68.4-70.1 and 67.2-67.5), whereas realism-related dimensions are more sensitive: Video Realism decreases with lower resolution (95.8 to 94.0), and Audio Realism decreases with lower sample rate (47.5 to 45.3).
>
> Taken together, these results suggest that instruction-following evaluation is comparatively robust, while realism-oriented dimensions are more sensitive to signal-level degradation induced by configuration changes. We will clarify this trade-off in the revision: the benchmark preserves native outputs for realistic usage, but cross-system comparisons should be interpreted with awareness that realism scores may be affected by configuration differences.
>
> ## Q3:Unclear whether human annotator bias is controlled or quantified.
>
> R3: In our setup, annotator consistency is addressed at both the protocol and measurement levels. Before formal annotation, annotators received standardized training with detailed scoring rubrics, and we conducted intermediate quality checks during annotation to reduce careless or inconsistent labeling.
>
> To quantify reliability, we additionally computed system-level SRCC and PLCC (per model × dimension, averaged over 50 prompts; N = 20). The resulting agreement is high: human-human SRCC/PLCC = 0.853 / 0.886, while human-MLLM SRCC/PLCC = 0.816 / 0.864. These results suggest that human ratings are reasonably consistent and that the MLLM judge tracks human preferences closely at the system level.
>
> We agree, however, that annotator bias is not fully characterized by agreement statistics alone. In the revision, we will make both the annotation-control procedure and the reliability analysis more explicit.
>
> ## Q4: Ask for being open-sourced.
>
> R4: Yes. We plan to publicly release both the **benchmark code** and the **benchmark data/annotations**. At present, our intention is to adopt **separate licenses** for these two parts: the **codebase** will be released under a standard open-source software license (e.g., **MIT** or **Apache-2.0**), while the **benchmark data and annotations** will be released under a **non-commercial research license** such as **CC BY-NC 4.0**. We will clearly specify the final license terms and usage restrictions in the public release.

---

> > ### Author Rebuttal · Reviewer_YZPf · 2026-04-03
> >
> > Thank you for your response. I have a further question:
> >
> > The paper currently adopts several objective evaluation metrics based on foundation models. I would like to ask whether these models are sensitive to the input video frame rate or audio sampling rate, and whether such differences could affect the final evaluation results. For example, CLAP, which is used to measure consistency, typically requires audio input at 48 kHz as far as I know. In that case, if the generated audio from different models has different sampling rates, could this discrepancy influence the consistency evaluation results? From the perspective of fairness and sound evaluation practice, would it be more reasonable to first resample all audio to a unified sampling rate before conducting the evaluation?

---

> > > ### Author Response · Authors · 2026-04-04
> > >
> > > We thank the reviewer for this constructive follow-up question regarding evaluation fairness. To assess the practical impact of varying input formats on our objective metrics, we conducted a controlled sensitivity analysis using **100 videos**.
> > >
> > > **1. Audio Sampling-Rate Sensitivity:**
> > >
> > > We evaluated the impact of audio preprocessing on objective alignment metrics.
> > >
> > > | **Condition** | **T-A (CLAP)** | **A-V (ImageBind)** | **PQ** | **SQ** |
> > > | --- | --- | --- | --- | --- |
> > > | 48 KHz | 0.1799 | 0.2216 | 6.9315 | 1.9871 |
> > > | 16 KHz | 0.1803 | 0.2211 | 6.9330 | 1.9856 |
> > > | **Abs. change** | **0.0004** | **0.0005** | **0.0015** | **0.0015** |
> > >
> > > These results suggest that foundation model-based metrics like CLAP are highly robust to moderate audio preprocessing changes, yielding nearly identical scores.
> > >
> > > **2. Frame-Rate Sensitivity (Fixed 1080p):**
> > >
> > > We tested the effect of varying frame rates on the visual metric (DOVER). Here again, the changes are minimal and non-monotonic.
> > >
> > > | **FPS** | **VT (DOVER overall)** |
> > > | --- | --- |
> > > | 24 fps | 0.7543 |
> > > | 12 fps | 0.7519 |
> > > | 8 fps | 0.7551 |
> > >
> > > In a short, these controlled experiments suggest that moderate differences in native sampling rates or frame rates **do not** confound our main objective consistency/alignment conclusions. Nevertheless, we agree that unifying the formats (e.g., resampling to 48 kHz) is a more rigorous practice for standardized evaluation. We will integrate this unified preprocessing step into our open-source evaluation pipeline and include these sensitivity results in the appendix.

---

### Official Review · Reviewer_p7yL · 2026-02-27

**Soundness:** 3
**Presentation:** 3
**Significance:** 2
**Originality:** 2
**Overall Recommendation:** 3
**Confidence:** 3

**Summary:**

The paper introduces T2AV-Compass, a unified benchmark designed to evaluate Text-to-Audio-Video (T2AV) generation systems. The authors developed a dataset of 500 diverse and complex prompts using a taxonomy-driven pipeline and real-world video inversion. Furthermore, they proposed a dual-level evaluation framework that integrates objective signal-level metrics with a subjective protocol to assess instruction following and physical realism.

**Compliance With Llm Reviewing Policy:**

Affirmed.

**Final Justification:**

Thank you for your response and explanation. My score remains unchanged.

**Key Questions For Authors:**

1. Innovation & Depth: The paper is primarily a compilation of existing metrics with limited methodological innovation. It lacks a deeper analysis of how these results can concretely guide T2AV architectural improvements.
2. Metric Inaccuracy: Key objective metrics are outdated and lack discriminative power. Specifically, ImageBind and DeSync are insensitive to fine-grained temporal-semantic alignment, while Audiobox fails to accurately evaluate the quality and physical causality of complex "sounds."
3. Evaluation Bias: The subjective evaluation remains heavily dependent on the capabilities and inherent biases of the underlying MLLM (e.g., Gemini-2.5-Pro).

**Strengths And Weaknesses:**

1. Comprehensive Evaluation Taxonomy: Unlike existing benchmarks that focus on isolated modalities (video-only or audio-only), T2AV-Compass evaluates five key dimensions: video quality, audio quality, cross-modal alignment, instruction following, and realism.
2. High-Complexity Benchmark Dataset: The benchmark features 500 prompts with significantly higher token counts and semantic diversity compared to previous datasets. It specifically targets challenging scenarios like off-screen sounds and complex physical causality.
3. Dual-Level Framework (MLLM-as-a-Judge): The introduction of a structured subjective evaluation using MLLMs provides interpretable diagnostics through Q&A checklists, bridging the gap between low-level signal metrics and high-level semantic logic.
4. Large-Scale Benchmarking: The study provides a systematic comparison of 15 representative systems, including leading proprietary models (e.g., Veo-3.1, Sora-2, Kling-2.6), offering insights into the current state of the field.

---

> ### Author Rebuttal · Authors · 2026-03-31
>
> ## Q1: Lack deeper analysis of how these results guide T2AV architectural improvements
>
> **R:** Our contribution lies in the benchmark’s diagnostic depth rather than in proposing a single new metric. Current T2AV systems fail in multiple concrete ways that are often hidden by aggregate scores, including audio-video desynchronization, unrealistic timbre or acoustic texture, altered speech content, incorrect speaker/source attribution, and failures on long-horizon compositional prompts. T2AV-Compass is designed around these recurring failure modes, so that they can be explicitly observed and evaluated rather than collapsed into a single coarse score.
>
> For this reason, the benchmark is intended not only for ranking, but also for diagnosis and model improvement. By decomposing performance into alignment, instruction-following, and realism-oriented dimensions, our metrics help identify which capability is weak and therefore which model component or training data should be improved. For example, weak synchronization results may suggest the need for stronger temporal modeling, while failures in speech content or source attribution may point to the need for better speech supervision or source-aware training data.
>
> This also makes the benchmark useful for validating architectural changes. For instance, if a model improves instruction grounding by using stronger semantic features extracted from an MLLM, our IF-related metrics can directly test whether this architectural change leads to better prompt following. More broadly, the benchmark can help verify whether improvements in semantic conditioning, temporal modeling, or data coverage translate into gains on the specific failure dimensions they are intended to address.
>
> ## Q2: Metric Accuracy
>
> **R:** We agree that metrics such as ImageBind, DeSync, and Audiobox-derived audio-quality measures are not sufficient by themselves to capture fine-grained temporal-semantic alignment or physical sound grounding in T2AV generation.
>
> At the same time, we view these inherited metrics as useful but incomplete: they remain representative, stable, and reproducible objective measures for coarse semantic correspondence, synchronization, and unimodal fidelity. Our goal is therefore not to discard them, but to retain their strengths where they are reliable and supplement them where they are insufficient.
>
> This is why T2AV-Compass adopts a complementary framework: established objective metrics for signal-level properties, together with a taxonomy-driven MLLM-as-a-Judge layer for higher-level aspects such as instruction following, physical plausibility, material-timbre grounding, and complex audiovisual realism. We also note that Appendix D.2 / Figure 10 shows relatively weak correlations among most metrics, suggesting that they provide complementary rather than redundant signals.
>
> We agree that the current wording may overstate metric-level novelty. In the revision, we will clarify that our contribution is a unified diagnostic benchmark combining established objective metrics with judge-based evaluation, rather than a brand-new automatic metric for every T2AV challenge.
>
> ## Q3: Evaluation Bias with Gemini-2.5-Pro
>
> **R:** We agree that MLLM-based judging may inherit model-specific bias, and we do not treat Gemini-2.5-Pro as a bias-free oracle. The current submission therefore includes multiple validation analyses beyond a single judge output, including human-MLLM agreement (Appendix D.3), pairwise ranking validation (Appendix D.4), upper-bound calibration on real videos (Appendix D.5), and repeated-run robustness.
>
> We selected Gemini-2.5-Pro as the primary judge because, among the candidate MLLMs tested under identical settings, it achieved the best overall agreement with human ratings across the four subjective dimensions, although Audio Realism is its weakest dimension. As a robustness check, Gemini-2.5-Flash yields highly consistent rankings across the five evaluated systems (Spearman ρ = 0.9248), with unchanged relative ordering. This suggests that the main comparative findings are not tied to one specific Gemini variant alone.

---

> > ### Author Rebuttal · Reviewer_p7yL · 2026-04-03
> >
> > I appreciate the authors' response. My inquiry is intended to be a broader discussion on how this benchmark can effectively reveal current model limitations to guide future improvements. Given the variety of existing T2AV benchmarks, what is the core innovation of this work and its primary advantages over established benchmarks?

---

> > > ### Author Response · Authors · 2026-04-04
> > >
> > > Thanks the reviewer for raising this point. Allow summarize observations and the resulting design principles.
> > > Our empirical work has shown that, unlike LLMs, validation loss is not a reliable indicator of convergence when training video models. This necessitates the development of a dedicated benchmark for T2AV generation to monitor training progress and evaluate final model performance. Such benchmark must provide a comprehensive assessment, covering key aspects including audio-visual quality and the alignment between the generated content and the input text prompt. However, two major constraints exist: the high computational cost of video inference precludes extensive evaluation, and the financial cost of using LLM-based evaluators for rubric scoring is also substantial. These factors demand a benchmark that avoids exhaustive checklists for non-critical attributes.
> > > These considerations form the primary motivation behind the design of our benchmark, T2AV-Compass. When compared with two concurrent works:
> > > 1. VBench2.0: T2AV Compass does not require five inference runs for each of its 1000 prompts, making it significantly more practical for monitoring the pre-training of large video models. Furthermore, our approach avoids "metric-stacking". Instead, we focus on a concise set of the most critical metrics: audio-visual quality, audio-visual alignment, and text-prompt fidelity. This streamlined approach reduces evaluation complexity while effectively identifying a model's core deficiencies. Yet, it still provides a unified scoring system for ranking models and guiding the training process.
> > > 2. VABench uses relatively short prompts and QA-style completion metrics, which may provide limited diagnostic resolution for stronger T2AV systems. In our comparison, it yields a relatively compressed ranking in which Wan 2.5 appears close to stronger proprietary systems such as Sora and Veo. This is somewhat inconsistent with broader community perception, and contrasts with the clearer separation revealed by our benchmark under denser and more compositional instructions. More generally, VABench’s QA-style completion rates tend to cluster around 80%, which may limit sensitivity to capability differences among stronger models. By contrast, T2AV-Compass is designed as a fine-grained diagnostic benchmark, it provides clearer signals about where failure occur.
> > >
> > > Given that video model training is highly sensitive to data distribution and captioning styles, robust validation is essential. Unlike benchmarks where each sample tests only one or two narrow aspects, each prompt in T2AV-Compass is designed to assess multiple critical dimensions simultaneously. T2AV-Compass delivers an accurate and challenging evaluation across multiple dimensions, making it an effective tool for monitoring key metrics throughout the training of audio-visual generation models. We contend that such a benchmark is crucial for the effective pre-training of large-scale video models.
> > >
> > > We would also like to make the benchmark’s positioning relative to prior T2AV work more explicit. Our contribution is not primarily to introduce a brand-new low-level metric for every subproblem, but to build a benchmark that jointly improves prompt complexity, diagnostic granularity, and practical usefulness for model development. As shown in Table 1 and Fig. 1(b), T2AV-Compass is intentionally built around a much denser prompt distribution: our prompts average 154 tokens, compared with 65 in JavisBench and 50 in VABench, and they contain more subjects and events per prompt (4.03 / 3.61 versus 3.68 / 1.78 and 3.01 / 2.31). We believe this better reflects realistic T2AV usage, where models must handle long, compositional, and constraint-heavy instructions rather than short caption.
> > >
> > > We pair this dense prompt design with a structured evaluation framework that explicitly separates objective assessment from judge-based diagnosis. In particular, the instruction-following branch is organized into 7 primary and 17 sub-dimensions, while realism is retained as a separate first-class evaluation track rather than being folded into a single average score. This design makes T2AV-Compass useful not only for ranking systems, but also for diagnosing why they fail. Our empirical results show that current T2AV systems exhibit highly non-uniform weaknesses across dimensions: Tables 2–3 reveal a clear audio-realism bottleneck despite relatively stronger visual performance, while Fig. 9 shows that model performance degrades sharply as prompt complexity increases, especially for long-horizon event compositions.  We believe our results are especially valuable for model development, because they translate benchmark outcomes into actionable directions such as stronger semantic conditioning, better temporal modeling, and improved sound grounding.
> > >
> > > T2AV-Compass is designed not only to evaluate whether a model works, but to reveal which capability breaks first under dense and realistic T2AV instructions.

---

### Official Review · Reviewer_Xk8C · 2026-03-10

**Soundness:** 2
**Presentation:** 2
**Significance:** 3
**Originality:** 2
**Overall Recommendation:** 4
**Confidence:** 3

**Summary:**

This paper proposes T2AV-Compass, a benchmark for evaluating text-to-audio-video (T2AV) generation. Existing benchmarks mainly focus on unimodal or weakly multimodal settings. Although there have been some attempts to evaluate T2AV systems, they are often limited in terms of fine-grained taxonomies and evaluation metrics.  T2AV-Compass introduces 500 dense prompts and a unified evaluation framework that covers both objective metrics (video quality, audio quality, and cross-modal alignment) and subjective evaluation (instruction following and realism). Using this benchmark, the authors evaluate multiple T2AV systems and find that sound effects generation remains a major bottleneck in current T2AV models.

**Compliance With Llm Reviewing Policy:**

Affirmed.

**Final Justification:**

I appreciate the authors’ detailed rebuttal and comments, which resolved my concerns. My remaining concern is that the manuscript should be substantially revised to reflect this discussion. Currently, the comparison with VABench is somewhat misleading, and the unique challenges and useful guidance on evaluation metrics provided by this work are not clearly reflected in the submitted version. I encourage the authors to improve the clarity and strengthen the presentation in the revision. I have increased my scores.

**Key Questions For Authors:**

1. How does T2AV-Compass fundamentally differ from VABench? Since VABench also employs an MLLM-based judge and evaluates fine-grained audiovisual constraints using QA-style checks, it would be helpful if the authors clarified the key differences between the two benchmarks. In particular, are there specific design considerations in the prompt construction or evaluation metrics that are tailored specifically for T2AV generation?

2. The distinction between objective and subjective evaluation is somewhat unclear. The paper categorizes Instruction Following and Realism as subjective metrics, yet these are computed automatically using an MLLM-as-a-judge rather than human evaluation. Could the authors clarify the rationale behind this categorization and how it differs conceptually from the objective metrics?

3. The conceptual distinction between quality and realism is not entirely clear. For example, metrics such as Acoustic Artifacts Score appear closely related to audio quality. Could the authors clarify why realism is treated as a separate evaluation category rather than being included within quality metrics?

**Limitations:**

Yes

**Strengths And Weaknesses:**

### strength

1. The paper addresses the timely problem of evaluating text-to-audio-video (T2AV) generation, which is still relatively underexplored compared to unimodal generation benchmarks.

2. The prompts appear relatively complex, with longer average token lengths than those in prior benchmarks, suggesting that they are carefully designed.

3. The benchmark introduces Instruction Following (IF) and Realism as additional evaluation dimensions, aiming to capture fine-grained constraints and physical law beyond traditional metrics.

### weakness

1. Insufficient discussion of closely related benchmarks (e.g., VABench). The paper stated in the related work section ``existing benchmarks are often limited in fine-grained taxonomies and evaluation metrics. These limitations motivate the development of T2AV-Compass.''. However it is not very clear how the limitation is stated. However, it is not clearly explained how T2AV-Compass addresses these limitations compared to VABench. For example, VABench also uses an MLLM-based judge and evaluates fine-grained audio-visual constraints using QA-style checks and also check the physical plausibility. Additionally, Table 1 does not report IF and RE metrics for VABench, which makes the comparison difficult to interpret.

2. Many of the objective metrics are directly adopted from prior works (e.g., CLAP, ImageBind, DeSync) for similar evaluation purposes. While reusing established metrics is reasonable, the paper claims that current evaluation methods struggle to determine whether generated sounds correspond to visible events or whether multiple audio sources are synchronized with complex visual interactions. The newly introduced evaluation components also rely on MLLM-based judging rather than introducing task-specific metrics.

3. It is unclear whether these metrics adequately capture the unique challenges of T2AV tasks, such as fine-grained audiovisual interaction. In the example prompts shown in the appendix, multiple types of sounds are often present (e.g., i) action-related sounds and ii) background ambient sounds). In such cases, action-related sounds should correspond to visible events, while ambient sounds may not have direct visual counterparts. However, the current evaluation mainly measures audio-video alignment at a relatively coarse level (e.g., Audio-Video Alignment and Temporal Synchronization) or evaluates audio and video separately in both objective and subjective evaluation.

4. The paper aggregates Instruction Following (IF) and Realism into a single average score (Table 3) and says Veo-3.1 ranking first, although these dimensions capture conceptually different aspects of generation quality. It is unclear whether averaging them provides a meaningful overall metric.

#### minor weakness

1. L140: Figure H appears to be incorrectly referenced.

2. L138: It may be helpful to cite JavisBench and VABench again for clarity.

---

> ### Author Rebuttal · Authors · 2026-03-31
>
> ## Q1. Clarify the positioning relative to VABench
> R1: VABench is an important, closely related, and largely contemporaneous benchmark, but T2AV-Compass differs from it in both scope and design emphasis. First, VABench spans multiple settings, including T2AV, I2AV, and stereo audiovisual generation, whereas T2AV-Compass is deliberately focused on T2AV. This narrower scope lets us tailor prompt construction and evaluation more specifically to text-conditioned audiovisual generation.
>
> Second, T2AV-Compass is more prompt-centric and diagnostic in design. Compared with VABench, it uses denser, more compositional prompts with richer audiovisual constraints, and explicitly emphasizes challenging T2AV phenomena such as off-screen sounds, physical causality, long-horizon event structure, and cinematographic control.
>
> Third, although both benchmarks use MLLM-based judging and QA-style checks, the role of the judge differs. In T2AV-Compass, Instruction Following (IF) and Realism (RE) are treated as first-class diagnostic dimensions under a taxonomy-driven checklist, enabling more explicit failure diagnosis across visual, acoustic, and cross-modal factors.
>
> VABench is also a closely related contemporaneous work, we will revise the related-work discussion to make this comparison clearer.
> ## Q2. Clarify the novelty at the protocol level
> R2: Our motivation is that no existing objective metric alone is sufficient for diagnosing high-complexity T2AV generation. Existing metrics are still useful for signal-level properties such as semantic correspondence, synchronization, and unimodal quality, but they are insufficient for higher-level aspects such as prompt-conditioned instruction following, physical plausibility, and complex audiovisual realism.
> Our response is therefore a dual-level evaluation framework: inherited objective metrics for signal-level assessment, complemented by a taxonomy-driven MLLM-as-a-Judge protocol in which IF and Realism are treated as first-class diagnostic dimensions. Appendix D.2 / Figure 10 shows relatively weak correlations among most metrics, suggesting that they provide complementary rather than redundant signals.
> ## Q3. Discuss fine-grained sound-role interactions
> R3: Fine-grained sound-role interaction is a central challenge for T2AV evaluation, when prompts contain both action-linked sounds that should align with visible events and ambient sounds that may not have direct visual counterparts. Rather than ignoring this difficulty, we incorporate it explicitly into the benchmark design.
> The prompt taxonomy already covers these cases along two complementary dimensions, as summarized in Appendix Figure 12. Audio Spatial Composition distinguishes On-screen, Off-screen, and Mixed cases, while Audio Temporal Composition distinguishes Single, Sequential, and Simultaneous cases. As shown in Figure 12, Mixed prompts account for 55.6% of cases and Simultaneous prompts for 72.8%, indicating that a large portion of the benchmark is intentionally designed to cover the kind of complex audiovisual interaction raised by the reviewer.
> Our evaluation then probes these challenges through both objective and judge-based layers: coarse AV alignment and synchronization provide system-level signals, while checklist-based IF and Realism verify whether prompt-specific sound/event constraints are satisfied under complex multimodal instructions. In this sense, the benchmark goes beyond coarse alignment alone.
> We agree that the current version does not yet provide explicit source-level attribution for each sound role within a scene. We will clarify this scope limitation in the paper and highlight sound-role annotation and event-level grounding as important future directions.
> ## Q4. Clarify metric semantics and score reporting
> R4: Thank you for this helpful suggestion. We will clarify these semantics more carefully in the revision.
> - Objective vs. judge-based: objective refers to fixed-rule signal/model-based metrics, whereas IF and Realism are judge-based perceptual/semantic evaluations produced by the MLLM judge rather than human studies.
> - Quality vs. realism: quality mainly reflects signal fidelity, whereas realism captures perceptual plausibility and physical/commonsense consistency in context. An output may have relatively clean signals yet still feel unrealistic for the depicted scene.
> - Aggregation of IF and Realism: we agree the average should not be over-interpreted. We will revise the presentation so that IF and RE remain primary, and the average is shown only as a compact summary rather than a definitive overall metric.
> ## Q5. Fix minor clarity and citation issues
> R5: Thank you for catching these issues. We will fix the Figure H reference and add the suggested citations to JavisBench and VABench.

---

> > ### Author Rebuttal · Reviewer_Xk8C · 2026-04-02
> >
> > Thank you for the detailed rebuttal. I appreciate the authors' clarifications. However, several concerns remain regarding:
> >
> > * Q2. The motivation that no single objective metric can fully capture T2AV complexity is reasonable. However, the proposed dual-level evaluation framework appears to primarily combine existing objective metrics (e.g., CLAP, ImageBind, DeSync) with an MLLM-based judge. Additionally, the framework lacks guidance on how to interpret the interaction between the two levels, especially given that the metrics are intended to be complementary. For instance, if a model performs well on objective metrics but poorly on MLLM-based evaluation (or vice versa), it is unclear how one should assess overall model quality.
> > * Q3. I appreciate the clarification regarding prompt taxonomy and the inclusion of complex audiovisual scenarios. However, as the authors acknowledge, the benchmark still does not explicitly model or evaluate fine-grained sound-role attribution (e.g., distinguishing action-linked sounds from ambient sounds). If such distinctions are not explicitly captured at the evaluation level, it remains unclear what aspect of the benchmark is uniquely tailored to T2AV, beyond increasing prompt complexity. The current checklist-based evaluation appears to assess video and audio largely in isolation. In this case, it is unclear how this setup differs fundamentally from evaluating text-to-video and text-to-audio models separately using the same prompt and aggregating their results. This raises questions about whether the benchmark truly captures joint audiovisual interaction, which is central to T2AV.

---

> > > ### Author Response · Authors · 2026-04-04
> > >
> > > ## Q2. Clarify how to interpret the interaction between the two evaluation levels
> > >
> > > R2: We thank the reviewer for this important point. Our goal is not to collapse the two levels into a single notion of overall quality, but to use them as two complementary diagnostic views of T2AV performance.
> > >
> > > Objective metrics mainly capture signal-level properties, such as coarse semantic alignment, temporal synchronization, and unimodal fidelity. By contrast, the MLLM-based judge evaluates prompt-conditioned instruction following and perceptual realism under more complex audiovisual constraints. Disagreement between the two levels is therefore informative rather than noise. Strong objective but weak judge performance suggests that a model preserves coarse alignment while failing on fine-grained prompt requirements, physical plausibility, or realism. Strong judge but weaker objective performance suggests that outputs appear plausible overall while still showing weaker signal-level alignment or fidelity.
> > >
> > > We will revise the paper to make this interpretation explicit and present the two levels as a capability profile for diagnosis, rather than as interchangeable scores or a single automatic metric.
> > >
> > > ---
> > >
> > > ## Q3. Clarify what is uniquely tailored to T2AV beyond prompt complexity
> > >
> > > R3: We thank the reviewer for this important concern. Our benchmark is tailored to T2AV through both prompt construction and evaluation of cross-modal dependencies between visual events and sound roles.
> > >
> > > At the prompt level, the benchmark targets diverse and compositional audiovisual scenes rather than isolated text-to-audio or text-to-video generation. We audited all 500 prompts and found broad coverage of sound-role configurations:
> > >
> > > | Main category | Sub-categories (count / share) |
> > > | --- | --- |
> > > | Biological Sounds | Human Non-linguistic Sounds (116 / 23.2%), Animal Sounds (169 / 33.8%) |
> > > | Ambient Sounds | Natural (229 / 45.8%), Urban (73 / 14.6%), Living (137 / 27.4%), Virtual (61 / 12.2%), Industrial (44 / 8.8%) |
> > > | Mechanical Sounds | Rhythmic Sound (99 / 19.8%), Immediate Sound (183 / 36.6%), Physical Interaction (264 / 52.8%) |
> > > | Musical Sounds | Singing (18 / 3.6%), Music Performance (45 / 9.0%), Background Music (244 / 48.8%) |
> > > | Speech Sounds | Speech Sounds (196 / 39.2%) |
> > >
> > >
> > > A single prompt may contain multiple sound roles, so these categories overlap. These statistics show that the benchmark is designed to cover sound sources and roles that must be grounded in visual context, temporal events, and scene composition.
> > >
> > > These dependencies are also encoded in the structured JSON annotations in the supplementary material. For example, when a visible character is speaking, the speech signal is linked to that character rather than treated as generic audio. We will clarify this annotation scheme in the revision.
> > >
> > > At the evaluation level, we also explicitly test cross-modal reasoning. Among all 970 audio-related checklist questions, 307 (31.65%) require audio-visual joint reasoning:
> > >
> > > | **Audio-related checklist category** | **Count** | **Share** |
> > > | --- | --- | --- |
> > > | Pure-audio questions | 663 | 68.35% |
> > > | Audio-visual synergy questions | 307 | 31.65% |
> > > | Character-Speech Alignment | 101 | 10.41% |
> > > | Action-Sound Causality | 99 | 10.21% |
> > > | Environment-Visual Synergy | 44 | 4.54% |
> > > | Music-Visual Synergy | 7 | 0.72% |
> > > | Other AV-synergy questions | 56 | 5.77% |
> > >
> > > These questions test whether speech matches the visible speaker, whether actions trigger the correct sounds at the right time, and whether environmental or off-screen sounds are plausible given the scene. They cannot be reduced to separate T2V and T2A evaluation, because each modality may look plausible in isolation while their interaction is still wrong.
> > >
> > > More broadly, Appendix Figure 12 shows that Mixed audio-spatial composition accounts for 55.6% of prompts and Simultaneous audio-temporal composition for 72.8%, indicating that a substantial portion of the benchmark targets scenes with multiple sound roles and nontrivial audiovisual interactions.

---

### Official Review · Reviewer_19zc · 2026-03-13

**Soundness:** 2
**Presentation:** 3
**Significance:** 3
**Originality:** 2
**Overall Recommendation:** 4
**Confidence:** 5

**Summary:**

This paper introduces T2AV-Compass, a benchmark for evaluating text-to-audio-video generation. The benchmark contains 500 prompts built through a taxonomy-driven pipeline, and it evaluates systems using both objective metrics for video, audio, and cross-modal alignment, and subjective MLLM-based judgments for instruction following and realism. The paper reports results on 15 representative systems, including end-to-end proprietary/open models and composed pipelines, and highlights persistent weaknesses in audio realism, synchronization, and long-horizon compositional generation.

**Compliance With Llm Reviewing Policy:**

Affirmed.

**Final Justification:**

I thank the authors for the thorough rebuttal. My main concerns have been adequately addressed, and the additional analyses make the paper stronger and more convincing. I now find the paper to be a useful and timely benchmark contribution for T2AV evaluation.

That said, I still encourage the authors to strengthen the final version by more clearly articulating the benchmark’s core innovation relative to prior work, and by discussing the limitations of the MLLM-judge protocol more explicitly. Overall, I increase my score accordingly.

**Key Questions For Authors:**

Please refer to "Weaknesses" in "Strengths And Weaknesses". If the authors address the concerns I raised above, I would be willing to increase my score.

**Limitations:**

Yes

**Strengths And Weaknesses:**

# Strengths

1. The paper tackles an important evaluation gap in multimodal generation. T2AV systems are advancing quickly, and the community needs better benchmarks than relying on a collection of inherited unimodal metrics.

2. The evaluation metrics covered in the paper are relatively comprehensive. The combination of objective metrics is more useful than relying on a single automatic score.

3. The paper includes a broad empirical comparison across 15 systems, which gives the benchmark practical value and helps contextualize its contributions.

# Weaknesses
1. The pipeline in Figure 2 includes LLM rewriting and human refinement, but the paper does not show how much the prompts changed or how often human judges overruled the LLM.

2. In Section 4.2 (“Audio Realism Bottleneck”), the paper claims that “current T2AV systems appear to rely on shallow semantic tag‑matching rather than deep physical grounding.” It is unclear how this interpretation is supported by the experiments presented; please clarify how this claim is substantiated by your results.

3. It is unclear whether all four dimensions (IF Video, IF Audio, Video Realism, and Audio Realism) are measured using Gemini 2.5 Pro. Table 4 shows that Gemini‑2.5‑Pro performs noticeably worse in audio realism. This point is important because audio realism is one of the main findings of the paper.

4. The MLLM judge validation is too limited given how heavily the paper relies on it. The current Human–MLLM Judge Agreement Analysis is conducted only on IF Video, IF Audio, Video Realism, and Audio Realism. I recommend extending the Human–MLLM agreement analysis to cross‑modal alignment‑related metrics, as these are crucial for multimodal quality evaluation.

5. In the Human–MLLM Judge Agreement Analysis, the authors should also report SRCC and PLCC between model predictions and human subjective scores in the main result tables or in an appendix. Compared to L1 distance, SRCC and PLCC provide a more detailed view of the overall consistency and numerical mapping between model outputs and human ratings, which would improve the rigor and comparability of the experimental analysis

---

> ### Author Rebuttal · Authors · 2026-03-31
>
> ## Q1: How much did the prompts change, and how often did humans overrule the LLM?
>
> R1: In the prompt-rewrite branch, Gemini-2.5-Pro enriches missing audiovisual constraints. The average prompt length increases from 54 to 154 tokens, and the average number of constraint points from roughly 5 to 10, with more explicit acoustic events and cinematic conditions added. Thus, the change is not merely stylistic.
> Human refinement then removes non-compliant cases and prompts that are too long or too complex for a short video. The candidate pool is reduced from roughly 650 prompts to 400 retained prompts, showing that human verification is a substantive inclusion step rather than a spot check.
> In the video-inversion branch, Gemini-2.5-Pro first generates dense temporally aligned captions from real videos, after which human verification resolves discrepancies with the source content. This process yields 400 prompts from prompt rewriting and 100 from video inversion, for 500 final benchmark cases.
>
> ## Q2: Clarify how the “shallow semantic tag-matching vs. deep physical grounding” interpretation is supported by the results
>
> R2: We agree that the original wording in Section 4.2 was too strong. We therefore conducted a follow-up failure decomposition over Audio Spatial Composition (On-screen / Off-screen / Mixed), Audio Temporal Composition (Single / Sequential / Simultaneous), and prompt sound-type composition. Overall, the patterns are more suggestive of shallow semantic content matching than robust physical grounding, though we do not view this as a definitive causal attribution.
> First, failure tracks sound-category specificity more than simple physical complexity: Biological sounds are hardest (50.3%, +13.5pp), Musical easiest (40.5%, -4.3pp), while Mechanical is near average (42.9%, +0.3pp)). This suggests sensitivity to semantic category specificity rather than physical complexity alone.
> Second, Sequential prompts are harder than Simultaneous ones: 48.8% vs. 40.7% (+8.1pp) on average, with the same pattern in speech-free subsets (WAN2.6: +14.4pp; LTX2: +5.8pp; Sora2: +19.4pp). This is not fully explained by physical-grounding difficulty alone, since simultaneous mixtures are often acoustically complex, yet temporal ordering remains harder.
> Third, Off-screen prompts are often among the easiest despite lacking visible anchors. This category is dominated by Musical sounds (45%), has the fewest required audio elements (1.60), and under Sora 2 shows 18.0% failure versus 39.5% for On-screen prompts. This is also consistent with successes arising from generic or weakly grounded sound generation rather than tight binding to visible events.
> Taken together, these analyses suggest that an important audio bottleneck may lie in fine-grained semantic content matching and temporal control, not fully explained by physical grounding difficulty alone. We will revise the text accordingly and add a concise summary with supporting table/figure to the supplement.
>
> ## Q3: Gemini-2.5-Pro evaluation bias
>
> R3: We agree that MLLM-as-a-Judge may inherit model-specific bias, and we do not treat Gemini-2.5-Pro as a bias-free oracle. We selected it because, among the tested MLLMs under identical settings, it achieved the best overall agreement with human ratings across the four subjective dimensions, although Audio Realism remains its weakest dimension. As a robustness check, Gemini-2.5-Flash induces the same performance ordering across the five evaluated systems. We will clarify this limitation and the role of human-validation evidence more explicitly.
>
> ## Q4: Extend Human-MLLM agreement analysis to cross-modal alignment metrics
>
> R4: We agree that cross-modal alignment quality should also be validated against human judgment. However, IF-Video, IF-Audio, Video Realism, and Audio Realism are produced by the MLLM judge, so the appropriate validation is Human-MLLM agreement; alignment-related dimensions are measured by objective metrics, so the appropriate validation is human-metric correlation. We will make this distinction explicit in the revision.
> We added a human-rated cross-modal dimension, Audio-Video Synchronization, on the held-out subset. Five annotators rated whether generated audio occurs at the correct moments relative to key visible events on a 1-5 Likert scale. Human annotations are reasonably consistent (pairwise Spearman rho = 0.8660). Using the mean human score as reference, the corresponding synchronization metric achieves SRCC = 0.9172, indicating high consistency with human judgments of AV synchronization.
>
> ## Q5: Ask for Report SRCC and PLCC
>
> R5: We additionally compute SRCC and PLCC to complement our current L1-based disagreement and Bradley-Terry/Elo ranking analyses. On our annotated subset, Gemini-2.5-Pro vs. human ratings yields SRCC = 0.816 and PLCC = 0.864, while human-human agreement is 0.853 and 0.886. These results further support that the judge tracks human preferences closely at the system level.

---

> > ### Author Rebuttal · Reviewer_19zc · 2026-04-04
> >
> > I thank the authors for their rebuttal. The authors have appropriately addressed my concerns.
> > ﻿
> > As the discussion period is still ongoing, I will consider the final rating after taking into account the opinions of the other reviewers.

---

> > > ### Author Response · Authors · 2026-04-04
> > >
> > > Thank you for the kind follow-up and for indicating that our rebuttal has fully addressed your concerns. We are very grateful for your constructive feedback.
> > >
> > > As you monitor the ongoing discussions, please rest assured that we have also engaged deeply with the other reviewers, providing detailed clarifications and new experimental evidence to address all their follow-up questions.
> > >
> > > Since your initial concerns have now been completely mitigated, we would deeply appreciate it if you might consider adjusting your rating to reflect these positive updates.

---

### Decision · Program_Chairs · 2026-04-30

**Decision:**

Accept (regular)

**Comment:**

This paper presents a useful benchmark for text-to-audio-video generation, and reviewers agreed that the problem is important and that the empirical study is valuable. The initial concerns were about the paper’s positioning relative to related benchmarks such as VABench, the role and limitations of the MLLM judge, and how clearly the benchmark captures joint audiovisual interaction. The rebuttal addressed most of these points with stronger validation, clearer prompt statistics, a better explanation of the two evaluation levels, and added evidence that the benchmark includes substantial AV-synergy cases. Most reviewers were satisfied after the discussion, while one reviewer remained unconvinced. I have read the rebuttal and later discussion carefully and taken them into account in my decision. I think the paper is technically sound and the benchmark is useful, so I recommend acceptance. The final version should incorporate the main rebuttal clarifications directly into the paper, especially the comparison to related benchmarks and the limitations of the MLLM judge.